# Hydrodynamic Analysis of a Modular Integrated Floating Structure System Based on Dolphin-Fender Mooring

Nianxin Ren [1,2], Yuekai Yu [2], Xiang Li [1,*] and Jinping Ou [3]

1   School of Civil and Architectural Engineering, Hainan University, Haikou 570228, China
2   State Key Laboratory of Coast and Offshore Engineering, Dalian University of Technology, Dalian 116024, China
3   School of Civil and Environmental Engineering, Harbin Institute of Technology, Shenzhen 518055, China
*   Correspondence: lixiang2180@163.com

**Abstract:** For both the expansion of important islands/reefs and the development of marine resources in South China Sea, a modular integrated floating structure (MIFS) system with tidal self-adaptation dolphin-fender mooring (DFM) has been proposed. The DFM, coupled with wave energy converters (WEC), can serve as an anti-motion system. Considering both the modules' hydrodynamic interaction effect and the connectors' mechanical coupling effect, both dynamic responses of the MIFS system and the WEC's output power characteristics were investigated under typical sea conditions. Based on the comprehensive consideration of key factors (safety, economy, and comfort), the effects of both the DFM and module connectors were systematically studied for the MIFS system. Preliminarily optimal design parameters of corresponding connectors and WECs were suggested. The security of the MIFS system under extreme sea conditions was checked, and a promising survival strategy has been proposed. In addition, the modular expansion scheme of the MIFS system was further discussed, and the results indicated that the proposed MIFS system shows good expansibility. The WEC can not only improve both dynamic responses and the comfort of inner modules, but also make considerable wave energy contributions.

**Keywords:** modular integrated floating structure; dolphin-fender mooring; wave energy converter; modular expansion; tidal self-adaptation

## 1. Introduction

In recent decades, with the rapid growth of the population, the demands for renewable energy and developable lands around coastal cities have increased significantly. Coastal areas have become the focus of human settlements for a long time, and will attract more than 50% of the world's population by 2050 [1]. Therefore, the very large floating structure (VLFS), a kind of economical and eco-friendly technology, has attracted wide attention all over the world [2]. The structural forms of the VLFS can be generally divided into pontoon and semi-submersible types. The pontoon type, a kind of simple structure with convenient maintenance, is suitable for mild conditions. The semi-submersible type, a complex structure with better hydrodynamic behavior, can survive in harsh conditions [3]. Compared with land reclamation, the VLFS has the advantages of mobility, is less earthquake-affected, has less of an impact on the environment, and has a low cost [4]. Therefore, the VLFS is promising for practical applications.

The mooring system of the VLFS can be generally grouped into four main types: catenary mooring, single anchor leg mooring, tension-leg mooring, and dolphin-fender mooring (DFM). The DFM is mainly composed of a monopile and rubber fender, which is very effective for restraining the horizontal displacement of VLFS, allowing vertical displacement. Therefore, it is a good tidal adaptive mooring type for shallow water zones. Kim et al. [5] conducted numerical simulations and model tests to investigate the effects of key fender parameters on the dynamic characteristics of the DFM, and

potential applications of the DFM were suggested for the VLFS. Cho [6] developed a new wave load calculation method for the VLFS based on the DFM, which took the wave slamming loads (Fx) on the piles into consideration. The results indicated that the monopile mooring performs better than the multi-pile mooring in the view of the pile's maximum stress and deformation. Nguyen et al. [7] tried to integrate the DFM with mooring lines to reduce the hydroelastic responses of the VLFS, and some instructive findings are pointed out. Mohapatra et al. [8] investigated the hydroelastic response to oblique wave incidence on a floating plate with a submerged perforated base, and some instructive results were obtained.

Because of the huge size of the VLFS, the modularization design of the VLFS with flexible connectors seems promising for alleviating excessive internal connector loads, which is also conducive to its own construction, transportation, installation, expansion, and disassembly [4,9]. Compared with modules' relative motions with flexible connectors, the structural deformation of a single module can be ignored. Therefore, the rigid module flexible connector (RMFC) analysis method is more suitable for the preliminary conceptual design of the VLFS [10]. Wang et al. [11] conducted both numerical simulations and model tests to investigate the hydrodynamic interaction effect between two VLFS modules. The results indicated that the hydrodynamic interaction was sensitive to the wave period, and the longer wavelength tended to affect this more obviously. Xiang and Istrati [12] further pointed out that the hydrodynamic interaction and wave loads on marine platforms is not only affected by the wavelength, but by the relative ratio of the wavelength-to-structure length ($L_{wave}/L$).

Ding et al. [13] analyzed the dynamic responses of an eight-module VLFS, using both the program THAFTS-B and scale model tests, which provided a helpful reference for the optimal design of multi-module VLFS systems. Lu et al. [14] proposed a hinge connector for a modular VLFS conceptual system, and the results indicated that the flexible hinge connector can effectively reduce the connector forces of the VLFS. Ren et al. [15–17] tried to add the wave energy converter (WEC) power take-off (PTO) system to the outermost hinge connector to reduce motions of the outermost module of a modular VLFS system, as well as produce considerable wave energy. Wang et al. [18] proposed a new flexible connector for a modular VLFS system to reduce the connector loads by allowing the modules' relative pitch and surge to some degree.

The realization of the VLFS is usually limited by economic viability more than technical feasibility. Due to the high cost of initial investment, operation, and maintenance, very few VLFS systems have been deployed [19]. However, the scheme, combined with equipment using both offshore renewable energy and natural shelters, will increase the feasibility of the VLFS. Loukogeorgaki et al. [20] proposed a pontoon-type modular VLFS with floating breakwaters, and the effect of floating breakwaters on dynamic responses of the modular VLFS were clarified. Some researchers [21–24] have tried to couple additional floating plates with the WEC to reduce the dynamic responses of modular VLFS systems and generate enough electricity to reduce VLFS operational costs. Nguyen et al. [25] designed a two-mode WEC attached to the up-wave end of a VLFS. Cheng et al. [26] addressed a VLFS with moonpool and an array of an oscillating buoy type WEC. The moonpool between two pontoons can improve wave energy extraction. So far, there are limited research studies on the tidal adaptive mooring system of the modular VLFS, especially for shallow water zones.

In this work, a novel expandable modular integrated floating structure (MIFS) system with tidal self-adaptive dolphin-fender mooring (DFM) is proposed for shallow water zones. Considering both the modules' hydrodynamic interaction effect and the connectors' mechanical coupling effect, the main hydrodynamic responses characteristics of the proposed MIFS system were systematically investigated. The effects of different connector types, WEC's key parameters and the modular expansibility were clarified. In addition, the security of the MIFS system under extreme sea conditions was checked, and a promising survival strategy was suggested.

## 2. Numerical Model of the MIFS

A novel expandable modular integrated floating structure (MIFS) system with tidal self-adaptive dolphin-fender mooring (DFM) was proposed for a certain mild sea zone, which is designed to work near artificial or natural shelters (islands or reefs) with a water depth of 20 m in the South China Sea [15,16]. The tidal self-adaptive DFM is easy to deploy and expand, and especially suitable for shallow water zones with significant tidal effects. The WECs have been embedded into the modules' connectors and the DFM, and can take advantage of modules' relative motions to generate electricity, as well as improve the dynamic responses of the MIFS system.

### 2.1. Description of the MIFS System

The conceptual sketch of the MIFS system is presented in Figure 1, which mainly includes:

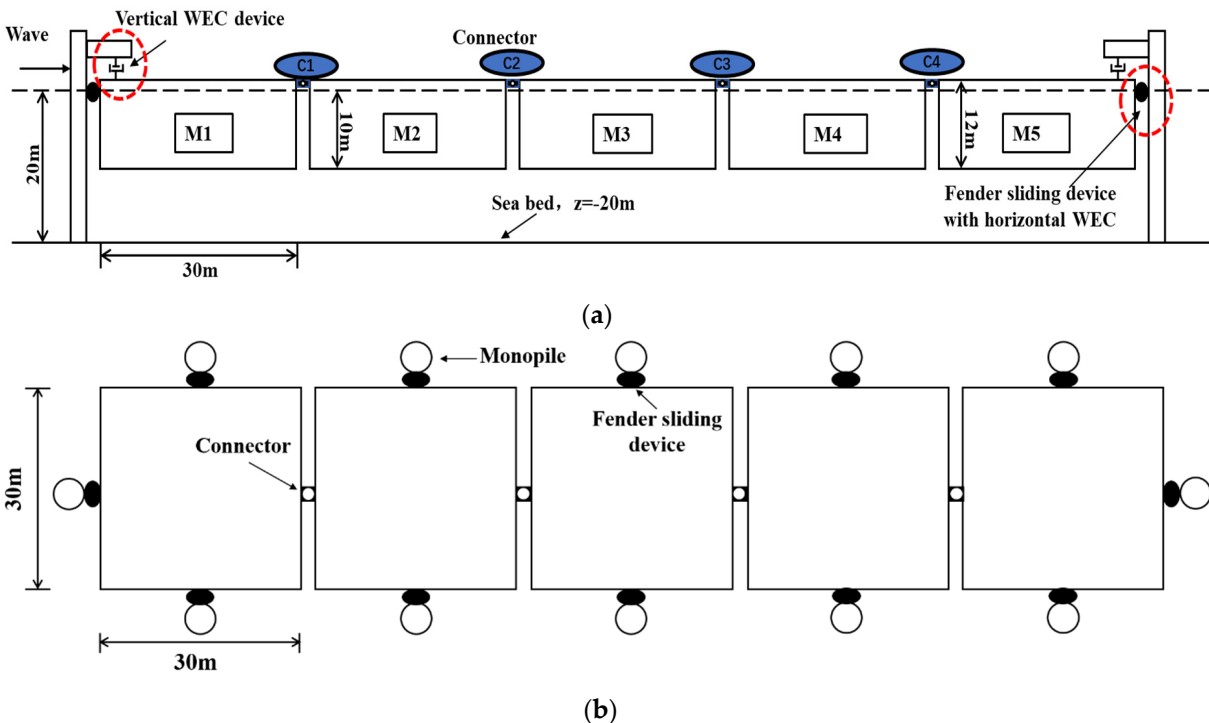

**Figure 1.** Conceptual sketch of the MIFS with DFM: (**a**) side view; and (**b**) top view.

(1)　Pontoon-type floating module

The gap among adjacent modules is suggested to be 2 m;

(2)　Tidal self-adaptive dolphin-fender mooring (DFM)

It consists of the fender sliding device, the vertical restraint device and the monopile. The fender sliding device can allow the heave motion (along the monopile) of the module but limits the module's horizontal motion with the effect of a linear spring. The vertical restraint device can serve as a vertical "damper" to mitigate the relative heave motion between the module and the monopile;

(3)　Module connector

Three typical types of module connectors have been taken into consideration, which are listed as follows [16]:

(a)　The hinge connector (denoted as "Hinge"): is only free for the relative pitch motion between the two connected modules;

(b)　The hinge connector coupled with an additional WEC (denoted as "HWK"): the PTO system of the WEC has been simplified as a linear pitch damper, which can



effectively mitigate the relative pitch motion of the two adjacent modules, as well as generate power;

(c)     The fixed connector (denoted as "Fixed"): There is no relative motion in all degrees of freedom between the two connected modules;

(4)     Wave energy converter (WEC).

It is installed between the module and the monopile, can capture wave energy by using both relative horizontal and vertical motions between the module and monopile, and effectively limit the motions of the module (as well as connection loads acting on the monopile).

The main structural design parameters of the MIFS system are shown in Table 1.

**Table 1.** Main structure design parameters for the MIFS.

| Parameters | Value | Units |
|---|---|---|
| Single module size | $30 \times 30 \times 12$ | m |
| Draft; Operating water depth | 10; 20 | m |
| Height of center of gravity | $-5.47$ | m |
| Mass = Displacement | 9225 | t |
| $I_{xx} = I_{yy}, I_{zz}$ | $1.08 \times 10^9; 1.46 \times 10^9$ | kg·m$^2$ |

*2.2. Multi-Body Dynamic Coupling Model*

Considering that the structural deformation of the proposed MIFS mainly occurs in connectors among adjacent modules, the module can be simplified as rigid body with flexible connectors. Thus, the government equation of the MIFS system can be generally summarized as follows:

$$M_i \ddot{X}_i + C_i \dot{X}_i + K_i X_i = F_{i,Wave} + F_{i,Con} + F_{i,\text{Dol−Fen}} + F_{i,\text{fender}} \tag{1}$$

where $X_i$ (6 Degree of Freedom, 6-DOF) indicates the generalized displacement vector of the *i*-th module, $\dot{X}_i$ and $\ddot{X}_i$ are the first and the second derivatives of the generalized coordinate vector to time, respectively. $M_i$, $C_i$ and $K_i$ denote the mass matrix, the damping matrix, and the hydrostatic restoring matrix, respectively. $C_i$ is an artificial damping commonly used to compensate the viscous fluid effect. $F_{i,Wave}$, $F_{i,Con}$, $F_{i,\text{Dol−Fen}}$ and $F_{i,\text{fender}}$ are the generalized wave force matrix, the connector force matrix, the dolphin-fender force matrix, and the possible bottom fender impact force matrix, respectively. The subscript number *i* (*i* = 1~5) of each matrix indicates the *i*-th standardized module along the incident wave direction.

The dolphin-fender force on the *i*-th module can be calculated as follows:

$$F_{i,\text{Dol−Fen}} = Kd_i X_i \tag{2}$$

where $Kd_i$ is the corresponding equivalent stiffness coefficient of the dolphin-fender for the *i*-th module.

The connector forces ($F_{i,Con}$) acting on the *i*-th module induced by adjacent modules can be expressed as follows:

$$F_{i,Con} = \sum_{j=1}^{5} \left( \varphi_{ij} Kc_{ij} \delta(X_i, X_j) \right) \tag{3}$$

where $\varphi_{ij}$ is a topology matrix. The $\varphi_{ij}$ is set to be one when the *j*-th module connects with the *i*-th module, otherwise $\varphi_{ij}$ is set to be zero. $Kc_{ij}$ is the connection stiffness matrix between the *i*-th module and the *j*-th module. $\delta(X_i, X_j)$ is the relative motion matrix between the *i*-th module and the *j*-th module.



The anti-collision fender has been equipped at the bottom of the two outermost modules, which can be used to monitor the potential bottom impact force. The bottom fender impact force $F_{i,\text{fender}}$ can be simplified estimated as follows:

$$F_{i,\text{fender}} = \begin{cases} Kf_{ij} \cdot \delta x(X_i, X_j) & \text{if } \delta x(X_i, X_j) < -2 \text{ m(contact)} \\ 0 & \text{if } \delta x(X_i, X_j) \geq -2 \text{ m(no contact)} \end{cases} \tag{4}$$

where $Kf_{ij}$ ($1.0 \times 10^7$ N/m) is the bottom fender linear stiffness coefficient between the $i$-th module and the adjacent $j$-th module (mainly for the two outermost modules). $\delta x(X_i, X_j)$ is the relative bottom surge motion between the $i$-th module and the adjacent $j$-th module. If the negative relative bottom surge motion $\delta x(X_i, X_j)$ is lower than the module's gap (2 m), the two adjacent modules will be bottom contacted, and the bottom fender contact force will be observed (in Figure 2).

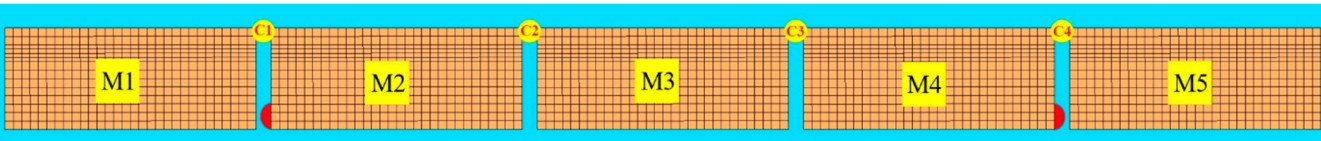

**Figure 2.** Hydrodynamic numerical model of the MIFS.

In addition, more detailed expression of the $F_{i,Wave}$, the $F_{i,Con}$ and the $F_{i,\text{fender}}$ can refer to the corresponding reference [14].

The output power $P_{i,WEC}(t)$ of the WEC coupled with the monopile can be calculated by corresponding WEC PTO damping force $F_{i,WEC}(t)$, the relative velocity $V_{i,WEC}(t)$ and the damping coefficient $K_{i,WEC}$:

$$P_{i,WEC} = F_{i,WEC}(t) \cdot V_{i,WEC}(t) = F_{i,WEC}(t)^2 / K_{i,WEC} \tag{5}$$

The output power $P_{i,Con}(t)$ of the WEC coupled with connectors can be calculated by corresponding WEC PTO damping moment $M_{i,Con}(t)$, relative angular velocity $\omega_{i,Con}(t)$ and the damping moment coefficient $K_p$:

$$P_{i,Con}(t) = M_{i,Con}(t) \cdot \omega_{i,Con}(t) = M_{i,Con}(t)^2 / K_p \tag{6}$$

*2.3. Hydrodynamic Model*

The program ANSYS-AQWA [27] is available for simulating the effects of shallow waves, the DFM, the second-order wave force, the connector's mechanical coupling, and the multi-body hydrodynamic interaction [1,16,28]. Based on the ANSYS-AQWA, a time-domain coupled numerical model of the proposed MIFS system with the DFM has been established, and the corresponding hydrodynamic numerical model is shown in Figure 2. Modules and connectors are marked as the $M_i$ ($i$ = 1–5) and the $C_i$ ($i$ = 1–4) respectively.

Five modules are involved in the hydrodynamic model, so the hydrodynamic interaction effect among modules must be considered. The total velocity potential can be generally written as follows:

$$\phi = \phi_I + \phi_D + \cdots + i\omega \sum_{i=1}^{5} \sum_{j=1}^{6} u_i^j \phi_i^j \tag{7}$$

where $\phi_I$ and $\phi_D$ indicate the potential of incident and diffraction, respectively. $u_i^j$ is the complex amplitude of the $i$-th module in the $j$-th modal (6-DOF). $\phi_i^j$ is the potential that is only caused by a unit amplitude motion of the $i$-th module, indicating the normalized velocity potential of the $j$-th modal of the $i$-th module.

## 3. Numerical Results of the 5-Module MIFS

Based on the established multi-body dynamic coupling time-domain numerical model of the 5-module MIFS system, the present work focuses on dynamic responses and WEC power characteristics of each module under typical sea conditions.

### 3.1. Deployment Effects of the DFM and Connectors

One certain selected sea area is of a water depth of 20 m with a natural islands-reef shelter effect [15,17], and one typical regular wave case is selected as follows: H = 4 m, T = 10 s. The preliminary design parameters of the DFM are as follows: the horizontal stiffness is $6.0 \times 10^6$ N/m with the horizontal WEC damping coefficient ($K_h$) is $8.0 \times 10^6$ Ns/m, the vertical WEC damping coefficient ($K_v$) is $6.0 \times 10^6$ Ns/m. The two outermost connectors (the C1 and the C4) are the HWK type, and the two inner connectors (the C2 and the C3) are the Fixed type. The preliminary design parameters of the HWK damping coefficient ($K_p$) is $1.0 \times 10^9$ Nms/rad.

The following three deployments of the MIFS are taken into consideration, and the corresponding dynamic responses are compared in Figure 3:

a. The $K_p$ is $1.0 \times 10^9$ Nms/rad, and each module is provided with vertical restraint devices;
b. The C1 and the C4 are changed into the Hinge type, and each module is provided with vertical restraint devices;
c. The $K_p$ is $1.0 \times 10^9$ Nms/rad, and the inner three modules (M2~M4) are without vertical constraint devices.

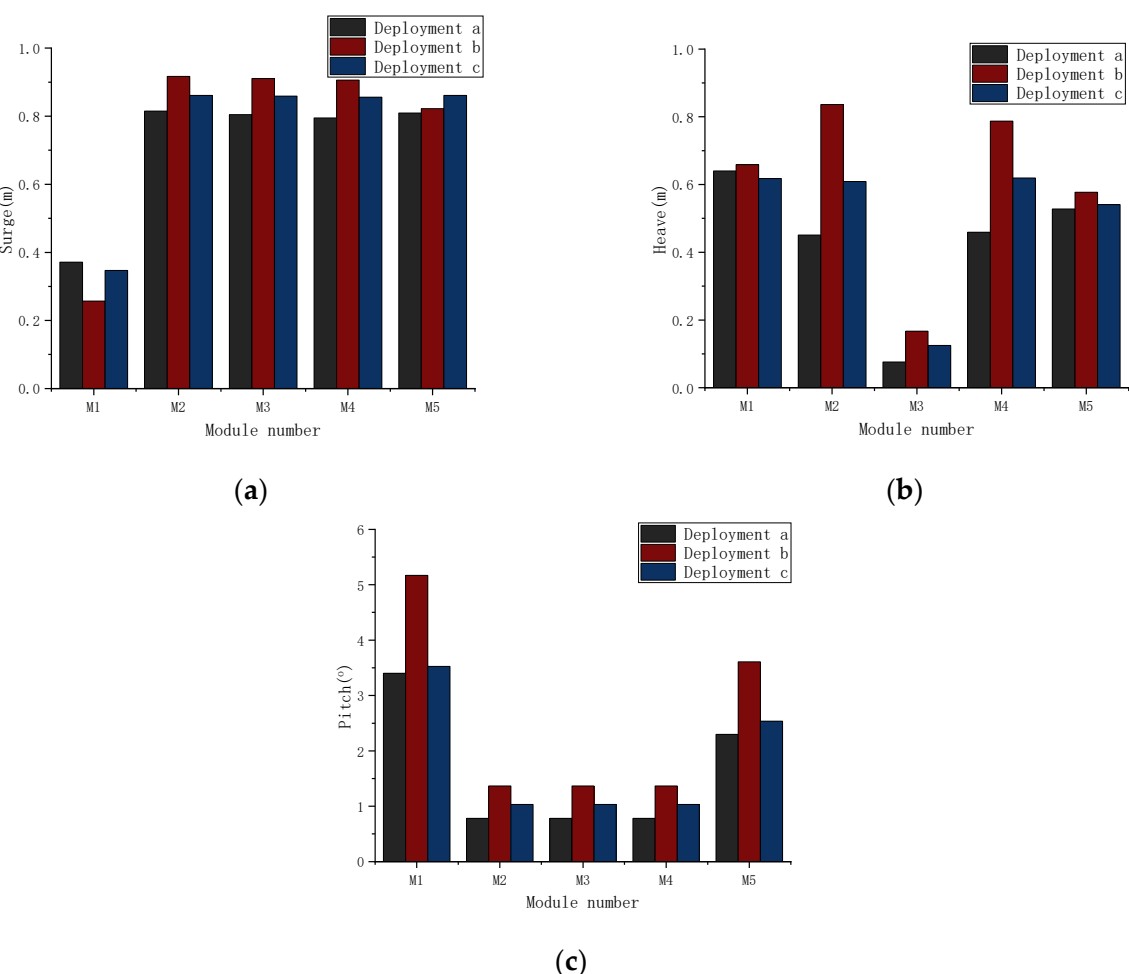

**Figure 3.** Dynamic responses of each module with three deployments: (**a**) surge; (**b**) heave; and (**c**) pitch.

In Figure 3a, the surge responses of the MIFS seem to not be very sensitive to different deployments. That is because the horizontal stiffness of the three deployments are almost the same, and the main difference of the three deployments are connecter-types and vertical restraint devices. The surge responses of the inner three modules (M2~M4) are much larger than those of the up-wave module (M1), especially for the deployment b. The surge of the up-wave module (M1) for the deployment b is the lowest among all modules.

In Figure 3b, different deployments have a great influence on heave responses of both the M2 and the M4, but less influence on other modules. The heave responses of the deployment b are the largest, while the heave responses of the deployment a are the smallest. That is because the hinge type of both the C1 and the C4 for the deployment b tends to induce lager pitch responses of all modules than the other two deployments, which can be seen in Figure 3c. Considering that the M2, the M3 and the M4 are fixed together, the larger pitch responses of the three inner modules can result in larger heave responses of both the M2 and the M4.

In Figure 3c, it can be seen that the pitch responses of deployment b are the largest among the three deployments, and the pitch responses of the three inner modules (M2~M4) are much smaller than those of the two outermost modules (M1, M5). A large relative pitch response between adjacent modules tends to result in terrible modules' bottom collision, which is very challenging for the safety of the MIFS system. Therefore, the deployment b is not recommended for the MIFS system. In addition, although the dynamic responses of the MIFS under deployment a is the smallest, the cost of the deployment a is the largest due to more additional vertical constraints. To balance both the safety and the economy, the deployment c is suggested for the MIFS system.

### 3.2. Effect of the HWK Key Parameter

Based on the deployment c, the optimal HWK design parameter ($K_p$) has been investigated under the typical sea condition (H = 4 m, T = 10 s). Four typical $K_p$ have been taken into consideration: ① $0.5 \times 10^9$ Nms/rad, ② $1.0 \times 10^9$ Nms/rad, ③ $1.5 \times 10^9$ Nms/rad, ④ $2.0 \times 10^9$ Nms/rad. The effect of different $K_p$ on main dynamic responses of the MIFS system has been clarified, and the results are shown in Figure 4.

In Figure 4a, the trend of the surge responses of the M1 with growing $K_p$ is opposite to that of the other four modules (M2~M5). The surge of the M1 is the smallest among all modules, and it increases with the increase of the $K_p$. However, the surge of the other four modules decreases as the $K_p$ increases. In Figure 4b, the heave responses of all modules decrease with growing $K_p$. The variation of the $K_p$ has a more significant influence on the heave responses of the inner three modules than those of the two outermost modules. Large $K_p$ can effectively reduce the heave responses of the inner three modules. The M3 is of the lowest heave among all modules, which is due to the fixed connector (C2 and C3) for the inner three modules. In Figure 4c, it can be seen that the pitch responses of the two outermost modules are much larger than those of the three inner modules, and the pitch responses of all modules decreases with growing $K_p$. More detailed information on the outermost HWK connectors is shown in Table 2. In Table 2, the maximum output power is 1.818 MW, when the $K_p$ is $1.0 \times 10^9$ Nms/rad. With the increase of $K_p$, the horizontal force ($F_x$) decreases, while both the vertical force ($F_z$) and the pitch moment ($M_y$) increase. Compared with the $F_x$ and the $F_z$, the $M_y$ is more sensitive to the variation of the $K_p$. Therefore, to balance both the safety and economic requirements, the $K_p$ is suggested to be $1.0 \times 10^9$ Nms/rad, which can generate more wave energy with acceptable connector loads.

### 3.3. Effects of Key Parameters of the DFM's WECs

Comprehensively considering safety, economy and comfort, the damping effect of the DFM's WECs on main dynamic responses of the MIFS system was further investigated under typical sea conditions (H = 4 m, T = 10 s). The horizontal damping coefficient ($K_h$) and the vertical damping coefficient ($K_v$) of the DFM's WECs are preliminarily set to be



$8.0 \times 10^6$ Ns/m and $6.0 \times 10^6$ Ns/m, respectively, and the corresponding results are shown in Figures 5 and 6.

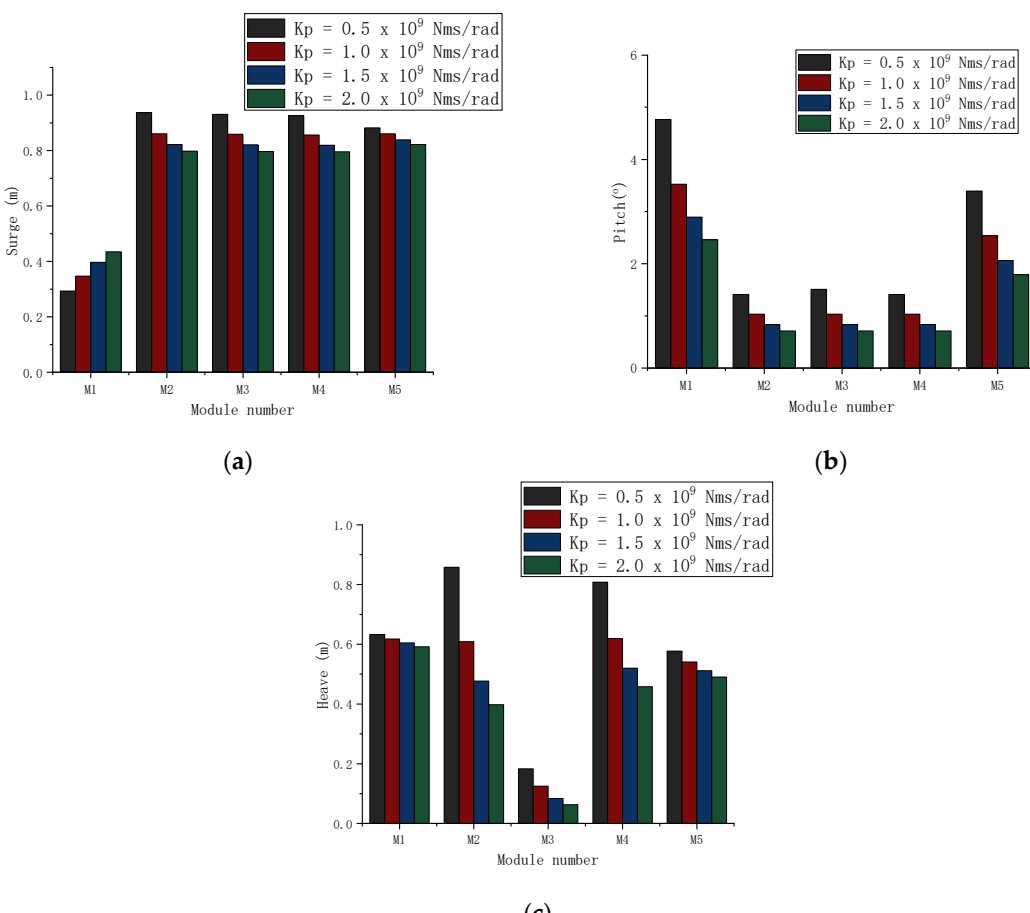

**Figure 4.** Dynamic responses of each module with different $K_p$: (**a**) surge; (**b**) heave; and (**c**) pitch.

**Table 2.** Mean output power and main C1 loads with different $K_p$.

| HWK Damping (Nms/rad) | Mean Output Power (MW) | Maximum C1 Load | | |
|---|---|---|---|---|
| | | $F_x$ (N) | $F_z$ (N) | $M_y$ (Nm) |
| $0.5 \times 10^9$ | 1.707 | $1.590 \times 10^7$ | $0.484 \times 10^7$ | $3.221 \times 10^7$ |
| $1.0 \times 10^9$ | 1.818 | $1.560 \times 10^7$ | $0.534 \times 10^7$ | $4.882 \times 10^7$ |
| $1.5 \times 10^9$ | 1.744 | $1.545 \times 10^7$ | $0.565 \times 10^7$ | $5.884 \times 10^7$ |
| $2.0 \times 10^9$ | 1.637 | $1.533 \times 10^7$ | $0.587 \times 10^7$ | $6.583 \times 10^7$ |

In Figure 5a,b, it can be seen that the horizontal damping ($K_h$) can effectively reduce both the surge and its acceleration of the whole MIFS system. Referring to the relevant acceleration comfort standards (NORDFORSK [29], the intellectual working comfort range, $a_h \leq 0.49$ m/s$^2$), it is suggested that the $K_h$ should be greater than $6.0 \times 10^6$ Ns/m. In Figure 5c, the output power of the horizontal WEC seems sensitive to the variation of the $K_h$, when the $K_h$ is less than $6.0 \times 10^6$ Ns/m. However, it becomes insensitive to the $K_h$, when the $K_h$ is larger than $6.0 \times 10^6$ Ns/m. For the horizontal WEC, the peaks of the mean output power of the M1 and the M5 appear at the $K_h$ of $6.0 \times 10^6$ Ns/m and $8.0 \times 10^6$ Ns/m, respectively. However, the peak of total horizontal WEC power appears at the $K_h$ of $7.0 \times 10^6$ Ns/m. Therefore, the optimal $K_h$ is suggested to be $7.0 \times 10^6$ Ns/m for optimal output power, as well as ensuring the safety and comfort of the MIFS system.

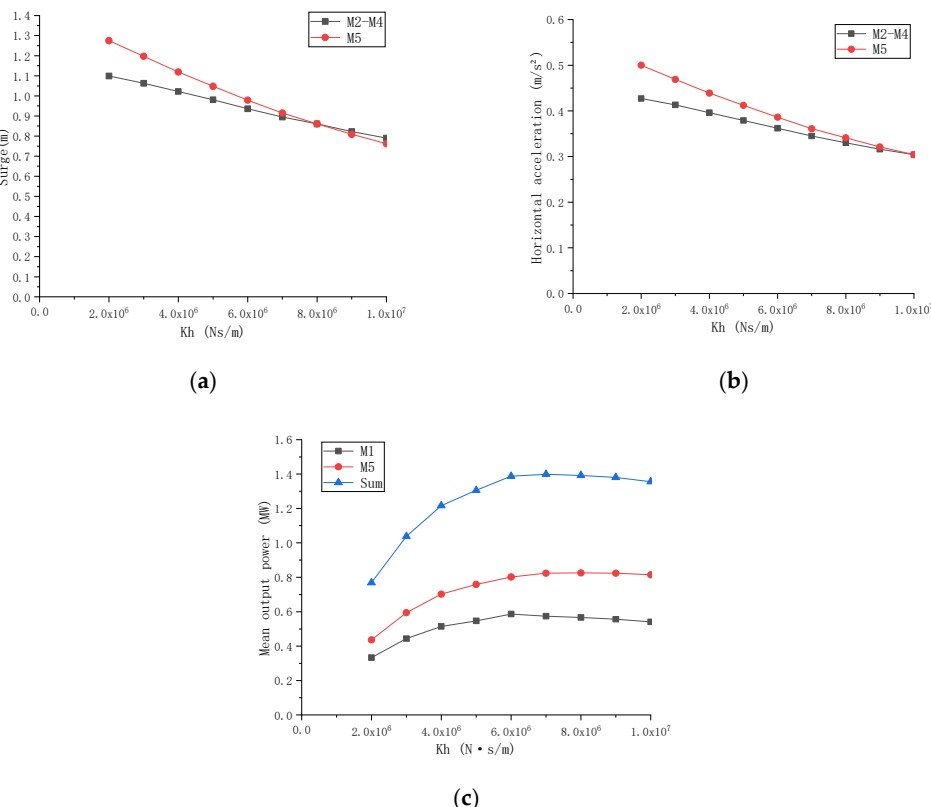

**Figure 5.** Horizontal dynamic responses and the mean output power with different $K_h$: (**a**) surge; (**b**) horizontal acceleration; and (**c**) mean output power.

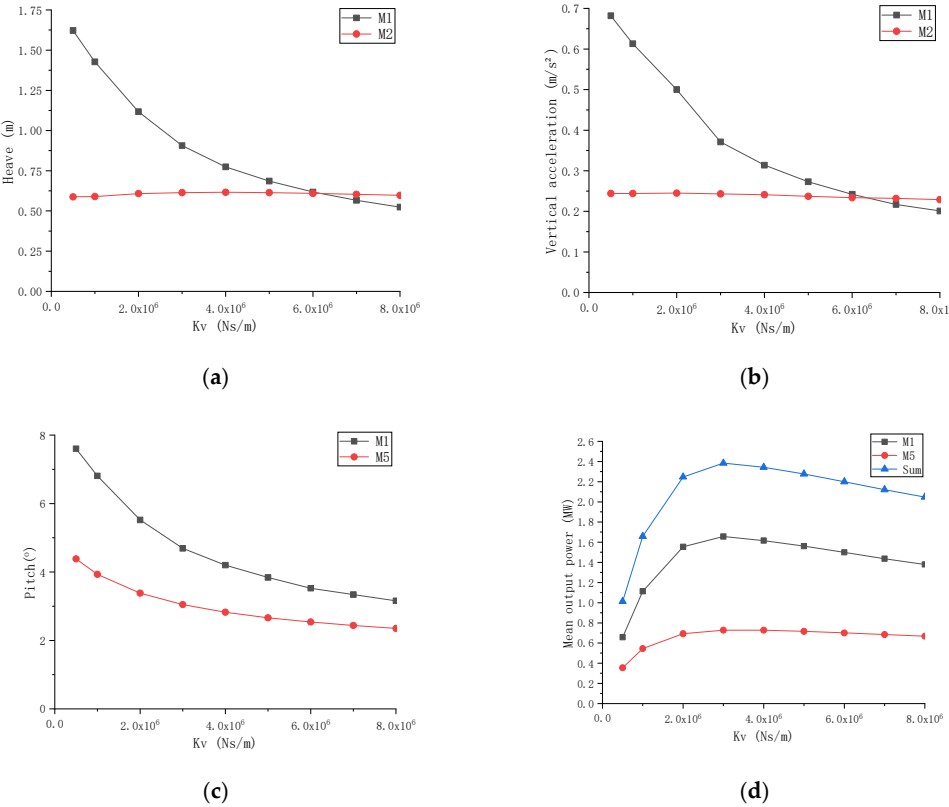

**Figure 6.** Vertical dynamic responses and the mean output power with different $K_v$: (**a**) heave; (**b**) vertical acceleration; (**c**) pitch; and (**d**) mean output power.

In Figure 6a,b, the $K_v$ has great influence on the heave responses of the outermost up-wave module (M1), while the M2 seems not sensitive to the $K_v$. Based on the reference to the relevant heave acceleration comfort standards (NORDFORSK [29], the intellectual working comfort range, $a_v \leq 0.98$ m/s$^2$), it is suggested that the $K_v$ should be greater than $3.0 \times 10^6$ Ns/m. In Figure 6c, the maximum pitch responses of the two outermost modules decrease with growing $K_v$, while the reduction rate decreases gradually. When the $K_v$ is less than $6.0 \times 10^6$ Ns/m, there is a risk of tbottom collision between the M1 and the M2, so the $K_v$ is suggested to be more than $6.0 \times 10^6$ Ns/m for the structure safety. In Figure 6d, the mean output power reaches the peak at the $K_v$ of $3.0 \times 10^6$ Ns/m, after that it gradually decreases. Therefore, based on the consideration of safety, comfort, and economy, the optimized $K_v$ is suggested to be $6.0 \times 10^6$ Ns/m. In summary, the preliminarily optimal design parameters of the MIFS system are listed in Table 3.

**Table 3.** Preliminarily optimal design parameters of the MIFS system.

| HWK | Fixed | Module with Vertical Restrain Device | $K_p$ (Nms/rad) | $K_h$ (Ns/m) | $K_v$ (Ns/m) |
|---|---|---|---|---|---|
| C1; C4 | C2; C3 | Only M1 and M5 | $1.0 \times 10^9$ | $7.0 \times 10^6$ | $6.0 \times 10^6$ |

### 3.4. Typical Operational Sea Conditions

A series of typical operational regular wave conditions (H = 2 m, T = 4~14 s) were considered, and the main dynamic response characteristics of the MIFS system under different wave periods were investigated. The corresponding numerical results are shown in Figure 7.

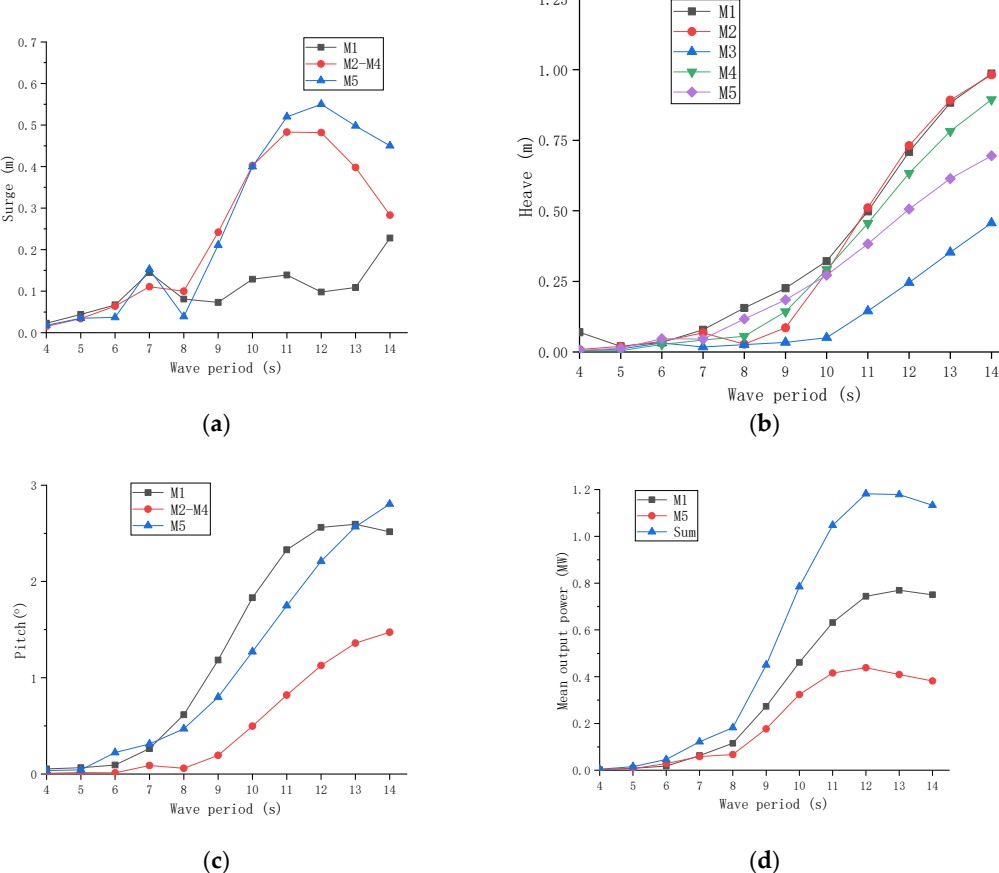

**Figure 7.** Main dynamic responses of the MIFS system under typical operational sea conditions: (**a**) surge; (**b**) heave; (**c**) Pitch; and (**d**) mean output power.

In Figure 7a, the surge responses of the up-wave M1 are generally smaller than other modules, especially for the wave period larger than 8 s. The surge responses of the modules M2–M5 have the same tend, reaching a peak at the period of 12 s. In Figure 7b, the heave responses of all modules increase as the wave period increases, especially for the wave period larger than 9 s. In Figure 7c, the pitch responses of the two outermost modules (M1 and M5) are much larger than those of the inner modules (M2–M4), and the pitch responses of the up-wave M1 reach the peak at wave period of 12 s. In Figure 7d, the mean output power of the two outermost modules both increase with the growing wave period, and reach the peak at the wave period of 12 s. The output power of the up-wave M1 is much larger than that of the down-wave module M5, and the power generation of the MIFS system seems considerable for the wave period larger than 8 s.

For the wave period smaller than 9 s, main dynamic responses of the MIFS system are not significant. Due to the shielding effects of the outermost module M1, both the heave and pitch of the inner modules are relatively low, which can be used for more functional designs. Considering the wave period of the selected zone with the water depth of 20 m is usually lower than 9 s, the MIFS system is of acceptable performance.

### 3.5. Extreme Sea Condition

For certain practical engineering, the MIFS system may be usually designed for working with the artificial or natural shelters (islands or reefs) in a mild sea zone, the wave condition for the MIFS system may be of unidirectional character according to "the entrance direction" of the shelters. The incident wave angle of $0°$ may be the most favourite direction for the MIFS system. Therefore, the paper mainly focused on the wave angle of $0°$. Considering the practical sea is always complex with random features, one representative extreme irregular wave condition was considered for the JONSWAP spectrum ($\gamma = 3.3$, $H_s = 4$ m, $T_p = 9$ s) [14]. Extreme dynamic responses characteristics of the MIFS system under the extreme sea condition were studied, and the main numerical results are shown in Table 4.

**Table 4.** Statistic information of main extreme responses of the MIFS system.

| Module | | Surge (m) | Heave (m) | Pitch (m) | Horizontal Acceleration (m/s$^2$) | Vertical Acceleration (m/s$^2$) | Horizontal Force of Monopile (N) | $M_y$ of Connector (Nm) |
|---|---|---|---|---|---|---|---|---|
| M1 | Max | 0.348 | 0.716 | 3.814 | 0.238 | 0.296 | $1.071 \times 10^7$ | |
| | Stdev | 0.274 | 0.284 | 2.168 | 0.131 | 0.170 | | |
| | | | | | | | | C1: $4.623 \times 10^7$ |
| M2 | Max | 0.840 | 0.599 | 1.042 | 0.379 | 0.257 | | |
| | Stdev | 0.601 | 0.211 | 0.408 | 0.214 | 0.067 | | |
| | | | | | | | | C2: $14.092 \times 10^7$ |
| M3 | Max | 0.840 | 0.168 | 1.042 | 0.379 | 0.007 | | |
| | Stdev | 0.601 | 0.007 | 0.408 | 0.214 | 0.023 | | |
| | | | | | | | | C3: $14.291 \times 10^7$ |
| M4 | Max | 0.840 | 0.617 | 1.042 | 0.379 | 0.234 | | |
| | Stdev | 0.601 | 0.245 | 0.408 | 0.214 | 0.113 | | |
| | | | | | | | | C4: $3.634 \times 10^7$ |
| M5 | Max | 0.869 | 0.551 | 2.684 | 0.385 | 0.259 | $1.146 \times 10^7$ | |
| | Stdev | 0.476 | 0.214 | 0.525 | 0.219 | 0.085 | | |

In Table 4, it can be seen that the extreme motion responses of each module are limited in a relatively stable range, and there is no observed bottom collision between the M1 and the M2. The horizontal and vertical acceleration of each module are also within the range of acceptable comfort. However, extreme connector's bending moment My on the two inner connectors (C2 and C3) can be over 140 MNm due to the fixed type, which are really challenging for the safety of the fixed connector. Therefore, the two inner fixed connector (C2 and C3) are tentatively changed into the HWK type ($K_p = 1.0 \times 10^9$ Nms/rad), to reduce the huge connector load. The main update results of the MIFS system with the new inner connectors are shown in Table 5.

**Table 5.** Statistic information of extreme responses of the MIFS system with releasing connectors.

| Module | | Surge (m) | Heave (m) | Pitch (m) | Horizontal Acceleration (m/s$^2$) | Vertical Acceleration (m/s$^2$) | Horizontal Force of Monopile (N) | My of Connector (N·m) |
|---|---|---|---|---|---|---|---|---|
| M1 | Max | 0.540 | 1.035 | 3.842 | 0.250 | 0.427 | $1.055 \times 10^7$ | |
|  | Stdev | 0.310 | 0.466 | 2.116 | 0.158 | 0.269 | | C1: $4.009 \times 10^7$ |
| M2 | Max | 0.515 | 0.907 | 3.019 | 0.256 | 0.358 | | |
|  | Stdev | 0.411 | 0.408 | 1.268 | 0.155 | 0.192 | | C2: $5.168 \times 10^7$ |
| M3 | Max | 1.049 | 0.897 | 3.089 | 0.502 | 0.374 | | |
|  | Stdev | 0.816 | 0.298 | 1.915 | 0.381 | 0.329 | | C3: $4.554 \times 10^7$ |
| M4 | Max | 0.991 | 0.823 | 2.415 | 0.425 | 0.332 | | |
|  | Stdev | 0.640 | 0.407 | 0.618 | 0.273 | 0.196 | | C4: $3.208 \times 10^7$ |
| M5 | Max | 0.736 | 0.775 | 2.295 | 0.335 | 0.363 | $1.114 \times 10^7$ | |
|  | Stdev | 0.426 | 0.308 | 0.978 | 0.186 | 0.125 | | |

In Table 5, after changing the two inner connectors from the Fixed to the HWK, the main motion responses of each module tend to increase to some degree, but they are still within the acceptable safety range (heavy manual work, the pitch motion $\leq 4°$). The comfort of the MIFS system can also meet the requirements of the heavy manual work ($a_h \leq 0.686$ m/s$^2$). The change of the monopile horizontal force seems neglectable, and the connector loads of the two inner connectors (C2 and C3) can be significantly reduced. Therefore, for extreme sea conditions, the Fixed inner connector can result in too-large connector loads, while changing the Fixed type into the HWK type may be a promising survival strategy for the MIFS system.

## 4. Modular Expansion Scheme Research

### 4.1. Modular Expansion Scheme

Considering that there may be a need for more modules of the original 5-module MIFS system in the long run, three possible modular expansion schemes have been considered, which are list as follows:

Scheme A: As shown in Figure 8a, seven modules are all deployed inside piles. The two expansion modules are connected on both ends of the original 5-module MIFS system, which are marked as the M6 and the M7, respectively. According to the deployment of the original DFM mooring system, there is a vertical restraint device on the tow outermost module (M6 and M7). The Fixed connectors are equipped among the three inner modules (M2–M4), and the HWK connectors ($K_p = 1.0 \times 10^9$ Nms/rad) are used for other connectors.

Scheme B: As shown in Figure 8b, two additional modules (M6 and M7) have been expanded symmetrically outside the monopile of the original 5-module MIF system. The two-expansion module (M6 and M7) are connected with the module M1 and the module M5 by the HWK ($K_p = 1.0 \times 10^9$ Nms/rad), respectively.

Scheme C: As shown in Figure 8c, the two expansion modules (M6 and M7) are fixed together, and the M7 is connected with the M1 by the HWK ($K_p = 1.0 \times 10^9$ Nms/rad).

Main dynamic responses characteristics of each module for the three expansion schemes under typical operating sea conditions ($\theta = 0°$, H = 2 m, T = 4~14 s) were investigated, and the corresponding results are shown in Figures 9–11.

The surge responses of the three expanded MIFS systems are shown in Figure 9. Three representative modules (M2, M5 and M6) are selected as the objects. Compared the Figure 9a with the Figure 7a, it indicates that the expansion Scheme A can effectively reduce the surge responses of the original MIFS system for wave period less than 11 s, which is due to the shield effect of more up-wave modules. However, it tends to result in larger surge for the wave period larger than 12 s, that is because the modules' shield effect decreasing with the increase of the wavelength, as well as more modules inducing more wave loads.

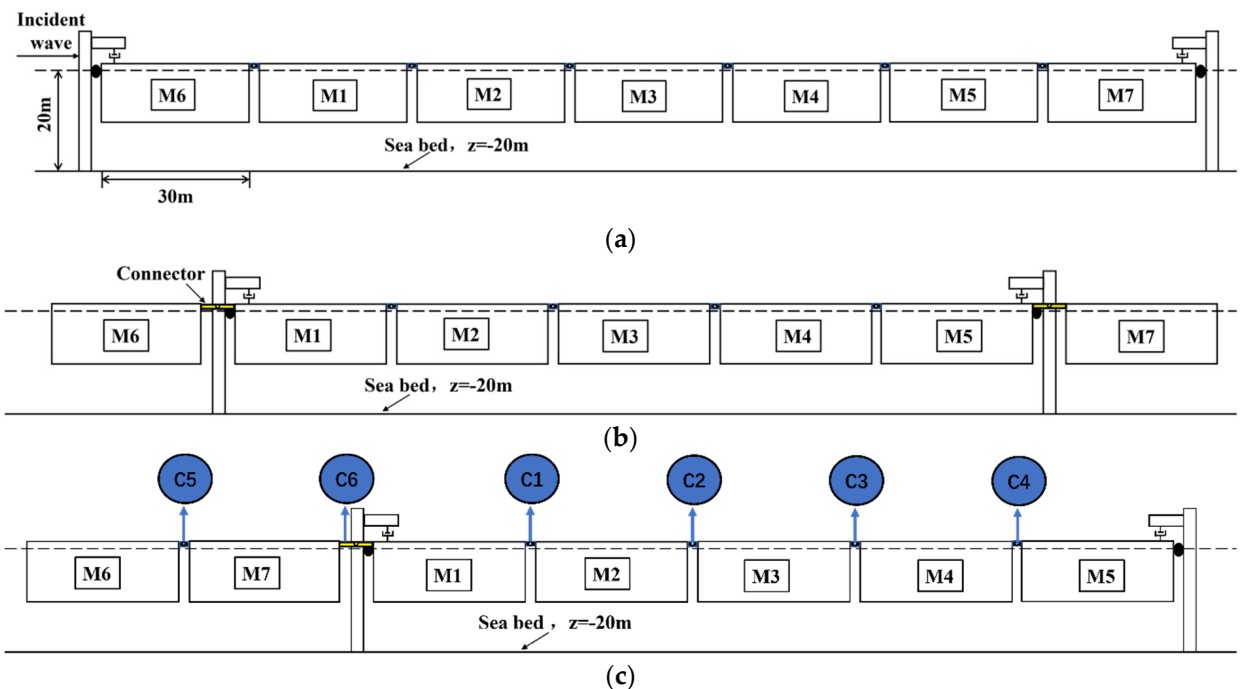

**Figure 8.** Three modular expansion schemes: (**a**) Scheme A; (**b**) Scheme B; (**c**) Scheme C.

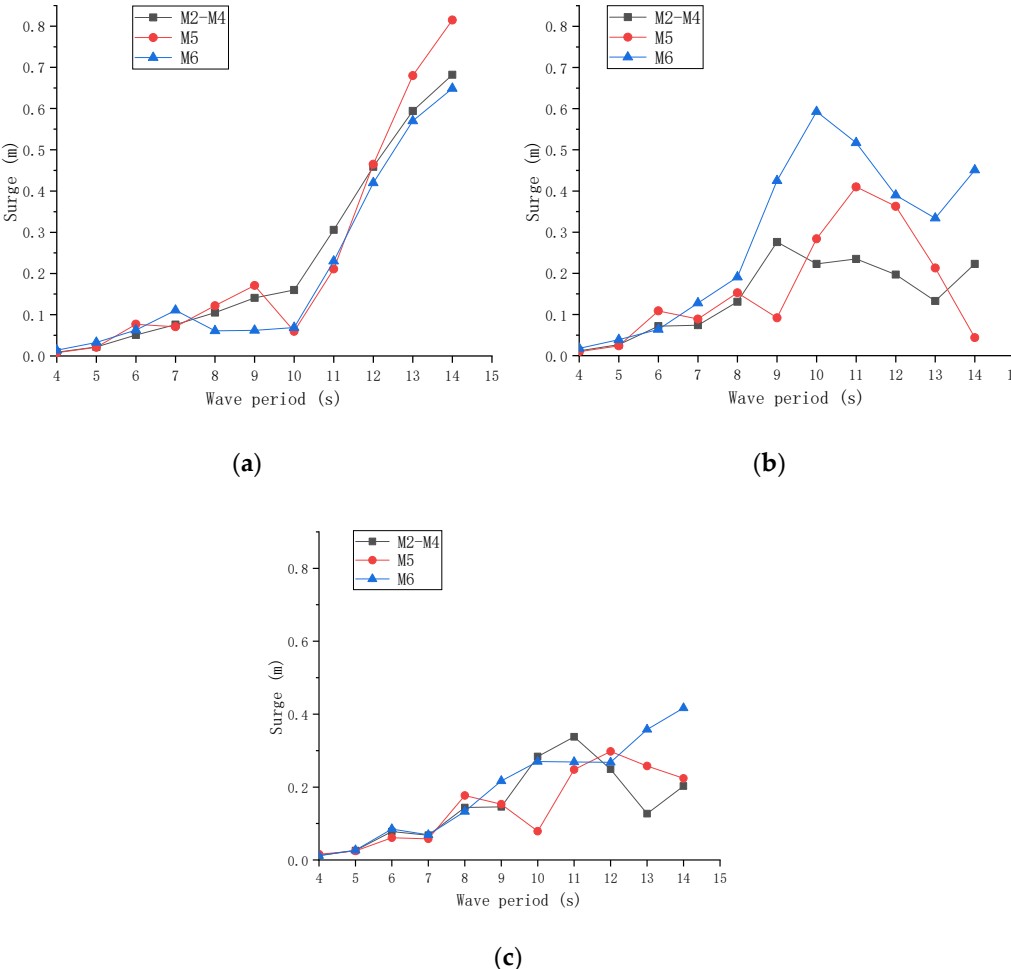

**Figure 9.** Surge of three expansion schemes: (**a**) Scheme A; (**b**) Scheme B; (**c**) Scheme C.

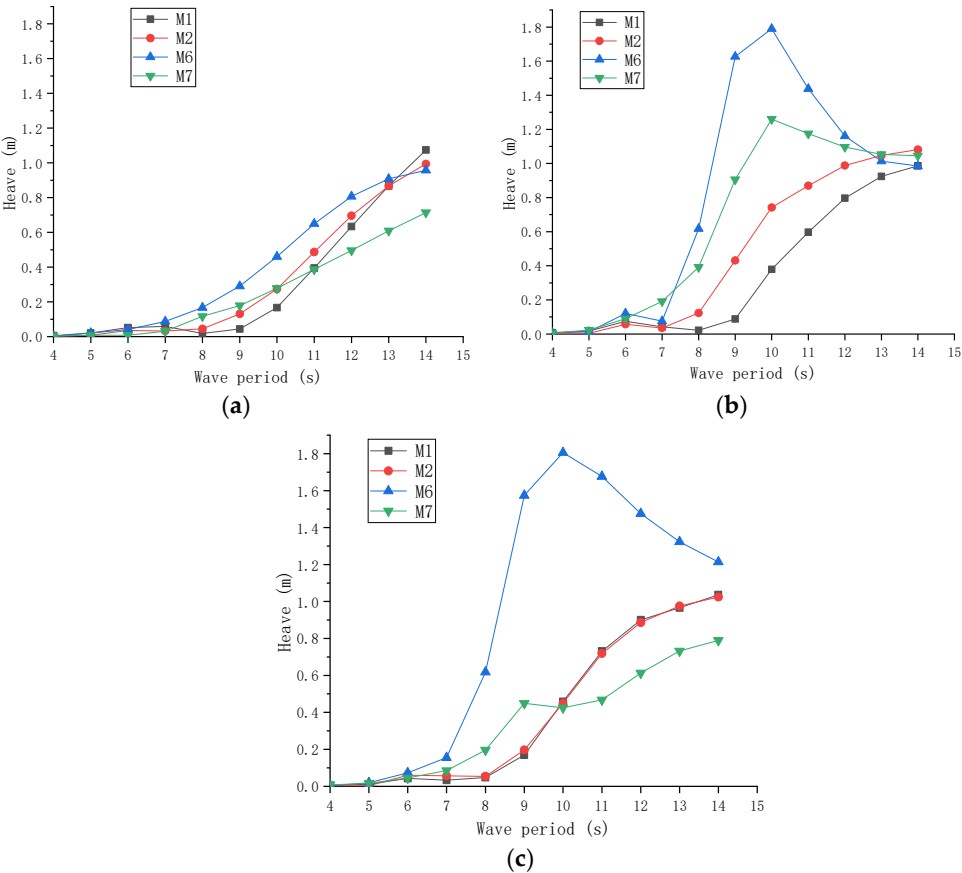

**Figure 10.** Heave of three expansion scheme: (**a**) scheme A; (**b**) scheme B; and (**c**) scheme C.

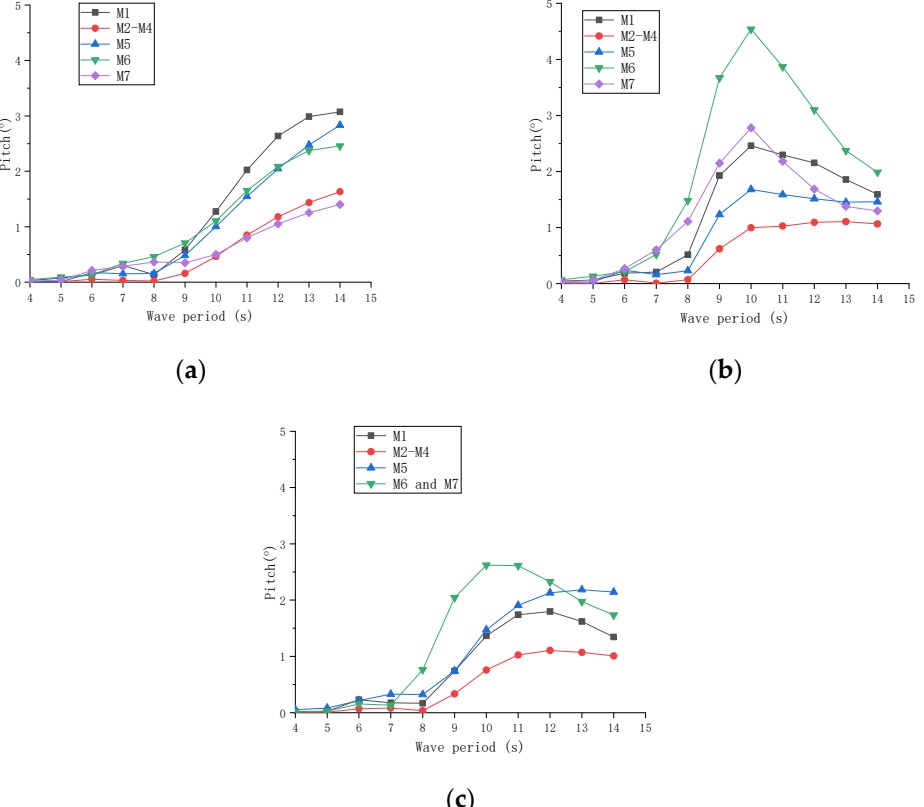

**Figure 11.** Pitch responses of three expansion scheme: (**a**) Scheme A; (**b**) Scheme B; (**c**) Scheme C.

Compared Figure 9b with Figure 7a, it can be seen that the surge trend of the expansion Scheme B is similar to the original MIFS system, while the surge responses of the inner modules (M2–M4) of the expansion Scheme B are much smaller than those of the original MIFS system. That's because that the additional M6 can play a breaking wave role for the original MIFS system.

Compared Figure 9c with Figure 7a, the surge responses of the expansion Scheme C are all much smaller than those of the original MIFS system, especially for the wave period larger than 9 s. That's because two additional modules play a more significant breaking wave role for the original MIFS system.

The heave responses of the three expanded MIFS systems are shown in Figure 10. Four representative modules (M1, M2, M6 and M7) are selected as the objects. Compared Figure 10 with Figure 7b, it can be seen that the heave responses of both the M1 and the M2 for the three expansion schemes is very similar with those of the original MIFS system, while the heave responses of the two additional modules (M6 and M7) for the three expansion schemes are different. That's mainly due to their different layouts. The two additional modules for both the scheme B and the scheme C are of more flexible feature than those for the scheme A, so it tends to induce larger heave responses. Therefore, it is necessary to pay attention to the heave responses of the additional modules, especially for potential large wave periods.

The pitch responses of the three expanded MIFS systems are shown in Figure 11. Five representative modules (M1, M2, M5, M6 and M7) are selected as the objects. Compared Figure 11 with Figure 7c, it can be seen that the pitch responses of the Scheme A is very similar with the original MIFS system, while the Scheme C with more significant wave-breaking effect can effectively reduce the pitch response of the original MIFS system. It should be noticed that pitch responses of the two additional modules (M6 and M7) for the Scheme B are much larger than those of the Scheme C, it tends to induce potential terrible bottom impact accidents between the M6 and the M1 for the Scheme B. in addition, representative transient responses (T = 11 s) of the three expansion schemes were compared in Figure 12, which may be helpful for a better understanding of their dynamic difference.

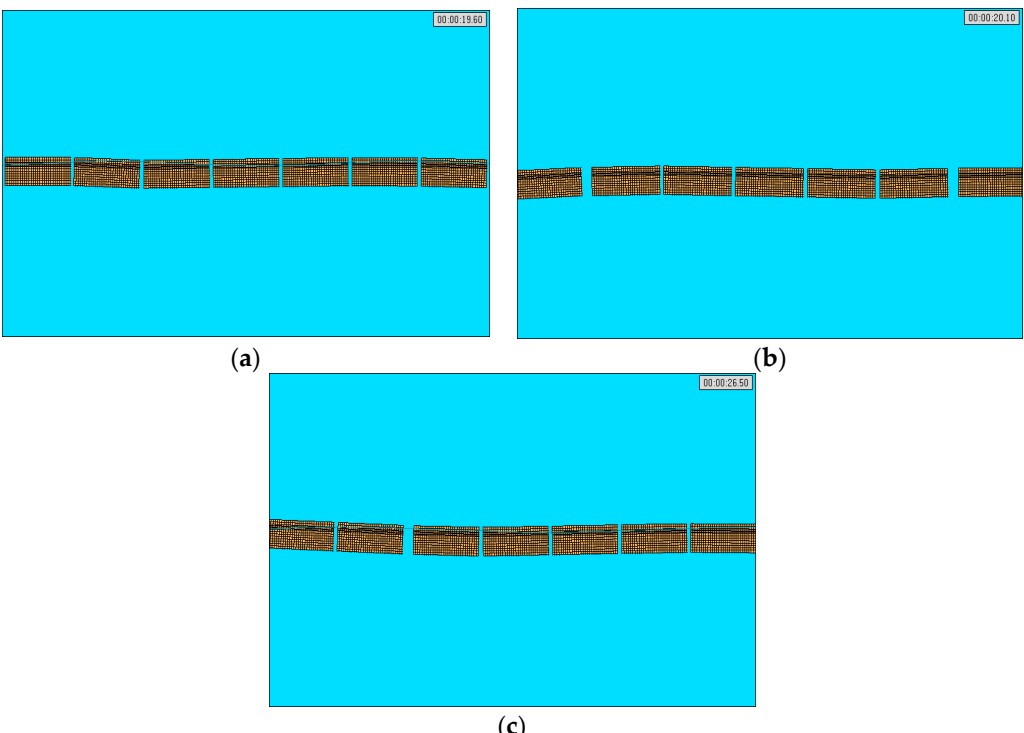

**Figure 12.** Representative transient responses of three expansion scheme (T = 11 s): (**a**) scheme A; (**b**) scheme B; and (**c**) scheme C.

Overall, the modular expansion tends to increase the characteristic period of the MIFS system. In the views of safety, convenience and economy, Scheme C is suggested for the promising expansion scheme for the MIFS system, which can appropriately sacrifice the dynamic responses of the additional expansion modules to ensure both the comfort and the safety of main inner modules.

### 4.2. Reverse Incident Wave Sea Conditions for the Scheme C

Because of the asymmetric deployment of the Scheme C, it is necessary to discuss the dynamic response characteristics of the Scheme C under 180° incident wave sea conditions (θ = 180°, H = 2 m, T = 4~14 s). Main dynamic responses results are shown in Figure 13.

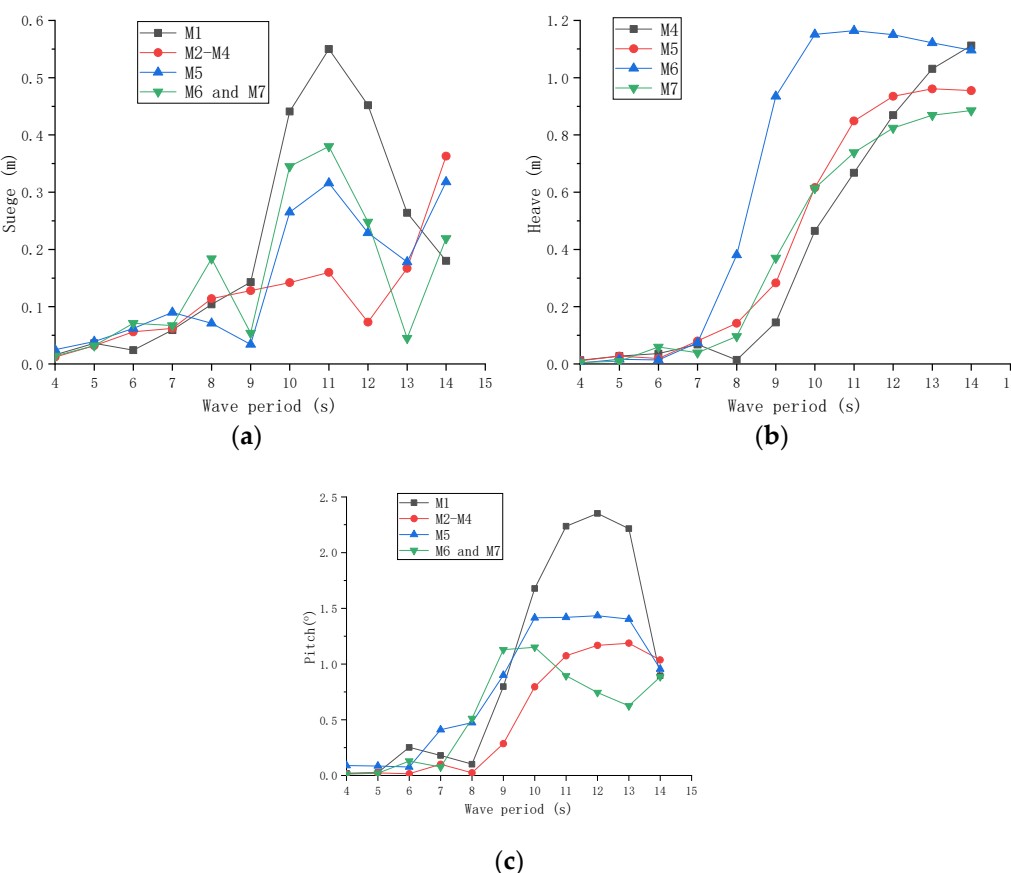

**Figure 13.** Numerical results of main motion responses of each module under 180° incident wave: (**a**) surge; (**b**) heave; and (**c**) pitch.

Compared Figure 13 with Figure 7, it can be seen that main motion responses of the Scheme C for the incident wave angle of 180° are very similar with those of the original MIFS system for the incident wave angle of 0° (in Figure 7a). However, the surge peak of the former appears at the wave period of about 11 s, which is slightly different from the later (12 s). That is because the two down-wave additional modules play less significant shield effect on the motion responses of the original MIFS system, compared with serving as up-wave wave breakers. Compared Figure 13b with Figure 10c, the heave responses of the two down-wave additional modules in Figure 12b are much smaller than those in Figure 10c. That is because of the positive shield effect of the modules in the original MIFS system. Compared Figure 13c with Figure 7c, it indicates that the two down-wave additional modules can also effectively reduce the pitch response of the MIFS system to some degree, and the pitch responses of the two additional modules are much smaller than those in Figure 11c. It is very beneficial for avoiding the potential bottom impact accident

between the M7 and the M1. Therefore, Scheme C is still feasible under the 180° incident wave sea conditions.

### 4.3. Extreme Responses of the Scheme C

Under the same irregular wave sea condition (JONSWAP, $\gamma$ = 3.3, $H_s$ = 4 m, $T_p$ = 9 s, incident wave angle of 0°), main dynamic responses of the expanded MIFS system with the Scheme C have been investigated, and both the C2 and the C3 are Fixed type. The corresponding statistic information of the expansion MIFS is shown in Table 6. The representative modules (M1, M2, M6 and M7) and connectors (C1, C2, C5 and C6) are selected as the objects. Comparing the results in Table 6 with those in Table 4, main motion responses (the surge, the heave, the pitch, the horizontal acceleration, and the vertical acceleration) of the inner module (M2) can be all effectively reduced to a considerable degree by the two additional up-wave modules, especially for the heave and the pitch. It is positive for improve human comfort. The horizontal forces of the monopile and connector loads are also reduced, especially for Fixed connectors. It can be beneficial for the safety of the MIFS system and makes the expanded deployment more reliable. However, it should be noticed that both the heave and the pitch responses of the two additional modules are much larger than those of the inner modules of the MIFS system, especially for the outermost M6. Therefore, if the two additional modules are used for potential storage function with limited human activities, the comfort level of which may be acceptable. In addition, there is no observed bottom impact force among modules, and the relative pitch between the M7 and the M1 can be used to generate considerable electricity for the MIFS system.

**Table 6.** Statistic information of the expanded MIFS system under the extreme sea condition.

| Module | | Surge (m) | Heave (m) | Pitch (m) | Horizontal Acceleration (m/s²) | Vertical Acceleration (m/s²) | Horizontal Force of Monopile (N) | $M_y$ of Connector (Nm) |
|---|---|---|---|---|---|---|---|---|
| M1 | Max | 0.498 | 0.610 | 2.102 | 0.285 | 0.269 | $0.956 \times 10^7$ | C1: $3.182 \times 10^7$ |
| | Stdev | 0.063 | 0.053 | 0.258 | 0.041 | 0.016 | | |
| M2 | Max | 0.487 | 0.689 | 0.949 | 0.239 | 0.310 | | C2: $10.482 \times 10^7$ |
| | Stdev | 0.006 | 0.033 | 0.015 | 0.016 | 0.031 | | |
| M6 | Max | 0.447 | 3.300 | 4.530 | 0.259 | 1.440 | | C5: $7.830 \times 10^7$ |
| | Stdev | 0.100 | 0.706 | 0.970 | 0.064 | 0.285 | | |
| M7 | Max | 0.447 | 0.873 | 4.530 | 0.259 | 0.401 | | C6: $5.836 \times 10^7$ |
| | Stdev | 0.100 | 0.164 | 0.970 | 0.064 | 0.030 | | |

### 5. Conclusions

In this work, a novel MIFS system based on DFM has been proposed, where the water depth is 20 m with natural shelters in South China Sea. Considering both the hydrodynamic interaction effect of modules and the mechanical coupling effect of connectors, main dynamic responses characteristics the MIFS under typical sea conditions have been investigated. Comprehensively considering the requirements of the safety, the comfort and the output power performance of the WEC, the effects of key design parameters of both the DFM and the module connector on dynamic responses of the MIFS system have been systematically investigated under typical sea cases, and the corresponding optimal parameters have been preliminarily suggested. Main conclusions are summarized as follows:

(1) When the wave period is large than 10 s, the global dynamic responses of the MIFS system increase obviously. However, considering the wave period for such shallow water (20 m) is usually smaller than 10 s, the hydrodynamic performance of the proposed MIFS system can be acceptable. Under extreme irregular sea conditions, the safety of the MIFS system has been checked. It should be noticed the fixed connector for the inner modules tends to suffer considerable large loads, which might be challenging for the safety of the MIFS system. Therefore, the HWK connector is

recommended to replace the inner fixed connector for reducing the huge connector loads, with sacrificing the modules' pitch responses to a certain degree. The results indicate that this strategy can effectively make the MIFS system safer;

(2) An effective modular expansion scheme has been proposed, which can improve the performance of inner modules. It indicates that the natural characteristic periods of the MIFS system tend to become larger with the modular expansion, which is more suitable for the shallow water environment. As a result, main dynamic responses of expanded MIFS systems are much better than those of the original MIFS system, especially for extreme irregular wave sea conditions. Therefore, the proposed MIFS system is of promising expansibility.

Many works still remain for the practical application of the proposed MIFS system, including the further optimization design, the validation of the scale model test, the soil-structure interaction effect of the monopole, multi-direction modular expansions, and the long-term performance of the MIFS system. The above-mentioned challenging works should be investigated in the near future.

**Author Contributions:** Conceptualization and Supervision: X.L. and J.O.; Investigation, methodology and writing: N.R. and Y.Y.; Funding acquisition: N.R. and X.L. All authors have read and agreed to the published version of the manuscript.

**Funding:** This research was supported by the National Natural Science Foundation of China (Grant No. 52161041), the Natural Science Foundation of Hainan Province (Grant NO. 520RC552, 520RC543, ZDYF2021GXJS034), Foundation of State Key Laboratory of Coastal and Offshore Engineering (Grant No. LP2119). The financial supports are greatly acknowledged.

**Institutional Review Board Statement:** The study was conducted in accordance with the Declaration, and approved by the Institutional Review Board.

**Informed Consent Statement:** Informed consent was obtained from all subjects involved in the study.

**Conflicts of Interest:** The authors declare no conflict of interest.

## Abbreviations

| | |
|---|---|
| DFM | Dolphin-Fender Mooring |
| DOF | Degree of Freedom |
| JONSWAP | Joint North Sea Wave Project |
| Fixed | Fixed Connector |
| $F_x$ | Horizontal Force of Connector |
| $K_p$ | Pitch Damping Coefficient |
| $F_z$ | Vertical Shear Force of Connector |
| $K_v$ | Vertical WEC Damping Coefficient |
| $K_h$ | Horizontal WEC Damping Coefficient |
| MIFS | Modular Integrated Floating Structure |
| My | Pitch Bending Moment of Connector |
| PTO | Power Take-Off |
| WEC | Wave Energy Converter |
| VLFS | Very Large Floating Structure |

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
