# Peer review of "Hydrodynamic Analysis of a Modular Integrated Floating Structure System Based on Dolphin-Fender Mooring"

_jmse, doi:10.3390/jmse10101470_

Round 1
Reviewer 1 Report
Dear authors, please see the attached file.

Author Response
Dear editors and reviewers:
We sincerely thank the editor and the reviewer for their valuable feedback, which are very helpful to improve the quality of our manuscript. The reviewer’ comments are laid out below in blue, and our responses are given in black. Revised portions are marked in red in the revised manuscript.
Response to the comments of reviewer #1:
General impression:
This study discusses the dynamic response of a modular floating structure system which is integrated with a WEC. Given (i) the increase of population in coastal areas and (ii) the global search for environmentally friendly energy, developing multi-purposes systems -such as the ones presented in the current study- that can solve both of these issues is important for society. Therefore, the reviewer believes that such a topic is worthy of investigation and fits the scope of the particular journal.
The manuscript is generally well written with only a few locations requiring further clarification, discussion or additional information. The reviewer provides specific comments below, which hopefully will help the authors address these weak points and improve the quality of the manuscript further before publication.
Response: Thank you for your positive comment!
Comments:
- L48-49: The authors mention here that Cho [6] developed a wave load calculation method for floating structures. So, what did this wave load calculation method include, e.g. did Cho [6] develop equations for the hydrodynamic pressures or for the horizontal and uplift forces (Fh and Fup)? Moreover, please note that in addition to the aforementioned method there are several other important studies that have developed simplified wave load calculation methods for marine platforms/decks (e.g. [n1]-[n4]) for the horizontal and uplift forces. In fact, [n4] developed equations for predicting not only the forces but also the large pitching moment, while [n5] developed simplified load cases (that included application of Fh and Fup at different locations of the structure) in order to represent the most critical instants of the wave impact and inundation of the marine structure.
Response: Thank you for this comment!
The responding description in this paper has been revised according to the comment.
- L55: Why do the flexible connections limit the internal force? Is this due to the dynamic fluid-structure interaction effect, which was seen in past studies (e.g. [n6], [n7]) to affect the forces that are applied on a platform or in connections of a deck? Can’t the flexible connections result in a dynamic amplification, as observed in past studies of wave impact on structures (e.g. [n8], [n9]), leading to larger forces in the connections of marine structures?
Response: Thank you for this comment!
Just as the reviewer said, the flexile connector may tend to amplify the relative motions among modules to some degree, but there is no wave impact effect observed in this work.
Based on our previous related work (the following reference), the flexible connectors can effectively reduce the internal connection loads, compared with the fixed connectors. That’s because the flexible connector can “sacrifice relative motions” among modules (to acceptable extends) to reduce the huge connector loads due to fixed connector, especially for the connectors’ bending moments (My). In the view of energy, the flexible connector can effectively transfer the wave loads (energy) into relative motions (kinetic energy) of adjacent modules, other than connectors’ structural deformation (internal energy). As a result, the internal forces (connector loads) can be effectively limited by proper flexible connectors. In addition, the wave impact effect hasn’t been considered in this paper. The corresponding description in this paper has been revised according to the comment.
- Ren, N.; Wu, H.; Ma, Z.; Ou, J., Hydrodynamic analysis of a novel modular floating structure system with central tension-leg platforms. Ships Offshore Struc. 2020, 15 (9), 1011-1022.
- Ren, N.; Wu, H.; Liu, K.; Zhou, D.; Ou, J., Hydrodynamic Analysis of a Modular Floating Structure with Tension-Leg Platforms and Wave Energy Converters. Mar. Sci. Eng. 2021, 9 (4), 424.
- Ren, N.; Zhang, C.; Magee, A. R.; Hellan, Ø.; Dai, J.; Ang, K. K., Hydrodynamic analysis of a modular multi-purpose floating structure system with different outermost connector types. Ocean Eng. 2019, 176, 158-168.
- L61-63: The authors mention here that the hydrodynamic interaction was sensitive to the wave period and longer wavelength tended to affect more obviously. Please note that a recent study [n10] demonstrated that the hydrodynamic interaction and wave loads on marine platforms is not only affected by the wavelength, but by the relative ratio of the wavelength-to-structure length (Lwave/L). In fact, the uplift forces remain constant after a certain value of the (Lwave/L), which would imply that the acceleration and response would not increase anymore. So, the authors are advised to provide an updated discussion in the introduction that will also highlight the importance of the non-dimensional ratio of Lwave/L.
Response: Thank you for this comment!
The corresponding description in this paper has been revised according to the comment, and the following reference has been added.
- Xiang T. and Istrati D. Assessment of Extreme Wave Impact on Coastal Decks with Different Geometries via the Arbitrary Lagrangian-Eulerian Method. Journal of Marine Science and Engineering, 2021, 9(12), 1342; https://doi.org/10.3390/jmse9121342
- Figure 1: Can you please show/draw on Figure 1 the location of the WECs? It is not clear from the description where they are located. Moreover, please show in this Figure also the connectors C1, C2, C3, C4, so that the reader is aware of their location before you mention them in lines 211-212.
Response: Thank you for this comment!
The responding description in this paper has been revised according to the comment, and more information has been added in Figure 1.
- Section 2.3: The authors mention here that they used ANSYS-AQWA for simulating the shallow wave. Many of past studies that used AQWA were based on the potential flow theory, which could cannot simulate wave breaking effects. So, did the authors investigate both unbroken and broken waves in the current manuscript? Given the fact that climate-change is expected to lead to more extreme storms and wave heights, it is expected that the probability of wave breaking to occur during extreme events will increase. Therefore, can the findings or at least the methodology used herein be expanded to breaking waves, broken waves or bores? For example, breaking/broken waves and bores are dominated by significant turbulence effects, as well as, air-entrapment during the breaking process (e.g. [n11]) and significant slamming loads and moments on marine structures, which are different from the one caused by unbroken waves (e.g. [n12]). Thus, the findings of the latter study would imply that they hydrodynamic response of the presented modular system in the current manuscript could also be different for unbroken and broken waves, both in terms of response (heave, surge, pitch etc) and consequently in terms of wave energy production. In the opinion of the reviewer it would be useful to provide some brief comments in the manuscript regarding the necessity to investigate such types of waves in future studies, the potential differences between unbroken and broken waves, the challenges that a researcher might come across when dealing with such more realistic wave types (e.g. difficulty to predict the high-frequency slamming loads, air-entrapment etc) and the feasibility (or limitations) of expanding the presented method/findings to such realistic scenarios.
Response: Thank you for this comment!
Just as the reviewer said, the breaking wave effect may be hard to be well considered in AQWA code. Considering the MIFS system usually designs to work near the artificial or natural shelters (islands or reefs) in a mild sea zone, the wave conditions for MIFS systems are not very serious (usually Hs< 4m). Therefore, the breaking wave effect may be not very significantly. That’s the reason why the breaking wave effect hasn’t been considered in this paper.
The AQWA code is widely-used for the hydrodynamic interaction analysis of modular floating systems. Some related references are listed as follows (our previous works), indicating that it is feasible and reliable for the hydrodynamic analysis of modular floating systems. Most of the numerical results based on AQWA are in good agreement with corresponding model test data. However, considering physical model tests are essential for validating the numerical results and improving numerical method, so we have been planning a model test at the end of this year. Then, we would like to compare the obtained test data with the corresponding numerical results, to prepare a new paper for submitting to the journal. Thanks for your kind consideration.
- Ren, N.X., Zhang, C., Magee, A. R., et al., 2019. Hydrodynamic analysis of a modular multi-purpose floating structure system with different outermost connector types. Ocean Engineering.176, 158-168.
- Ren, N.X., Wu, H., Ma, Z., et al., 2019. Hydrodynamic analysis of a novel modular floating structure system with central tension-leg platforms. Ships and Offshore Structures. Ships and Offshore Structures.15 (9), 1011-1022.
- Ren, N.X., Ma, Z., Shan, B.H., et al., 2020. Experimental and numerical study of dynamic responses of a new combined TLP type floating wind turbine and a wave energy converter under operational conditions. Renewable Energy. Ships and Offshore Structures. 151,966-974.
- Ren, N.X., et al., 2018. Experimental and numerical study of hydrodynamic responses of a new combined monopile wind turbine and a heave-type wave energy converter under typical operational conditions. Ocean Engineering. 159,1-8.
- Liu, Y.Q., Ren, N.X., Ou, J.P., 2021. Hydrodynamic analysis of a hybrid modular floating structure system and its expansibility. Ships and Offshore Structures. DOI:10.1080/17445302.2021.1996110.
- Yanwei Li, Nianxin Ren, Xiang Li*, Jinping Ou. Hydrodynamic Analysis of a novel modular floating structure system integrated with floating artificial reefs and wave energy converters. Journal of Marine Science and Engineering, 2022, 10, 1091. https://doi.org/10.3390/jmse10081091
- Section 3: The authors present here the effect of different parameters. Have they thought to investigate the effect of the in-plane dimensions of each module (which would affect the ratio Lwave/Lstructure that was seen to be very important in past studies (e.g.[n9])? For example, if you were to use 8 modules (with half of the original length of the 4 modules), how would the performance change?
Response: Thank you for this comment!
Just as the reviewer said, the modules’ dimension (or expansion) effect will affect the dynamic characteristics of the MIFS system to some degree (similar as previous mentioned Lwave/L effect). The corresponding findings have been reported in our previous works (list as follows), so more information can refer to following references.
- Liu, Y.Q., Ren, N.X., Ou, J.P., 2021. Hydrodynamic analysis of a hybrid modular floating structure system and its expansibility. Ships and Offshore Structures. DOI:10.1080/17445302.2021.1996110.
- Ren, N.X., Zhang, C., Magee, A. R., et al., 2019. Hydrodynamic analysis of a modular multi-purpose floating structure system with different outermost connector types. Ocean Engineering.176, 158-168.
- Ren, N.X., Wu, H., Ma, Z., et al., 2019. Hydrodynamic analysis of a novel modular floating structure system with central tension-leg platforms. Ships and Offshore Structures. Ships and Offshore Structures.15 (9),1011-1022.
- L208-210: How did the authors determine these initial values (Kh, damping coefficient, Kv). Did they select them based on prior studies of similar systems?
Response: Thank you for this comment!
Just as the reviewer said, the initial values of key parameters for the MIFS system are mainly based on our previous related work.
- L231: What is an “up-wave” module? The terms “up-wave” and “down-wave” do not seem to be the appropriate ones. Perhaps you should consider using the terms “first” and “last” module, the “offshore” and “onshore” module.
Response: Thank you for this comment!
The up-wave module denotes the first module suffering wave loads (the M1), and the down-wave module denotes the last module suffering wave loads (the M5). The corresponding description in this paper has been updated according to the comment.
- Figures 5-7: Can you please show the (i) connector forces, and (ii) monopile forces in Figures 5, 6 and 7? These two parameters are expected to be critical during the design process of the new system.
Response: Thank you for this comment!
The information of connector loads are mainly presented in Table 2, Table 4, and Table 5. Considering the connector loads in Figure 5-7 are not significantly different from those in Table 2, so the information of the connector loads is omitted. In addition, the information of connector extreme loads in Table 4 and Table 5 (extreme responses) may be more helpful for the safety design of the connectors.
- Section 4.1: The authors are advised to add some snapshots (from AQWA) of the wave-structure-interaction process for the three different Schemes (A, B, C) so that the reader can visualize the response of the structure (instead of explaining it only with words in the text). For example, you could select 3-4 critical time instants of the transient hydrodynamic response, e.g. when the wave interact with M6, when it moves forward and reaches M2, when it reaches M4 and when it reaches the last module (M7). This will make the article much more comprehensible to the reader.
Response: Thank you for this comment!
According to this comment, the comparison of representative transient responses of the three expansion Schemes has been added in this paper for a better understanding.
- (b)
(c)
Figure 12. Representative transient responses of three expansion scheme (T=11s): (a) Scheme A; (b) Scheme B; (c) Scheme C.
- L410-411: The authors mention here that more modules induce more wave loads. Since the loads on one module will probably be out-of-phase with the loads on other modules (i.e. the maximum load on each module most likely will NOT occur at the same instant), what would be the effect of the additional modules on the monopile forces? How do the forces in the connectors and monopiles (horizontal, vertical) change in Schemes A, B, C relative to the initial module?
Response: Thank you for this comment!
Just as the reviewer said, the expansion style of the original MIFS will have an effect on the dynamic responses of the MIFS system. We can compare the information between Table 4 and Table 6 (expansion with the proposed Scheme C), and it indicates that both the horizontal forces of the monopile and main connector loads (My) reduced, especially for Fixed connectors. That’s mainly due to the two additional outermost up-wave modules (in the Scheme C) serving as wave breakers for the inner modules. The corresponding description in this paper has been updated according to the comment.
- Line 456-457: The authors mention here that they investigated two wave angles (0 and 180 degrees). Why didn’t they investigate more angles in order to obtain a complete overview of the performance of the new system? These two wave angles seem to be an idealized scenario. In real life the wave direction can change. Moreover, previous research on different marine structures, such as low-crested structures, elevated decks and offshore platforms (e.g.[n13-n15]) has demonstrated a significant reduction of wave over-topping and of the wave-induced loads when there is a misalignment between the structure and the wave (e.g. oblique waves or skewed structures). In fact, the 3D effects generated in cases where the waves and the main axis of the structure are not aligned have a complex effect because they can reduce the hydrodynamic forces in one direction but generate additional out-of-plane forces, yaw and roll moments [n15], which should be taken into consideration in the design of any structure in the marine environment. Given the significant 3D effects demonstrated in the aforementioned studies, and the fact that recent work ([n16]) revealed that an extreme event (storm, hurricane) can generate different wave directions than the expected one by the analyses of local winds, the reviewer would advise the authors to review the aforementioned studies and run a few different relative angles between the wave and the VLFS (e.g. by changing either the wave direction or rotating the structural system), in order to see how the hydrodynamic performance of the system will change as a function of this relative angle. If this is not possible, the authors should at least discuss the importance of this relative angle based on available literature and clarify to the reader that future studies must investigate this parameter in order to develop really robust and optimized VLFS designs that can withstand any wave conditions and produce wave energy all year long, irrespective of the wave direction.
Response: Thank you for this comment!
Just as the reviewer said, the wave direction effect is important for better understanding of the MIFS system. However, considering the MIFS system may be usually designed for working with the artificial or natural shelters (islands or reefs) in a mild sea zone, the wave condition for the MIFS system may be of unidirectional character according to “the entrance direction” of the shelters. The incident wave angle of 0o may be the most favourite direction for the MIFS system. Therefore, the paper mainly focused on the wave angle of 0o. Based on our previous work, main dynamic responses of the MIFS system tends to increase with the wave angle increase from 0 o to 90 o. Therefore, the wave direction effect on the dynamic responses of the MIFS hasn’t been further investigated in this paper. It is highly suggested that the layout of the MIFS should well considered the unidirectional character of the wave condition. The corresponding description in this paper has been added according to the comment.
- Yaqiong Liu, Nianixn Ren, Jinping Ou. Hydrodynamic analysis of a hybrid modular floating structure system under different wave directions. Applied Ocean Research, 2022, 126: 103264.
- Conclusions: The first bullet point is not a conclusion. Therefore, it should be included in the first paragraph of the section, before the bullet points.
Response: Thank you for this comment!
The corresponding description in this paper has been revised according to the comment.
- General comment: What are the natural frequencies and modes of deformation (e.g. the first five frequencies) of the proposed systems? Can you present them in a new Table (for all the investigated schemes/configurations)? They could help explain further why the MIFS system interacts dynamically with some wave cases and shed light on the conditions that are critical (e.g. important wave periods/wave lengths that excite particular natural frequencies of the MIFS)
Response: Thank you for this comment!
Considering the MIFS system involving many modules with complex hydrodynamic and mechanical interaction effects, it may be hard to perfectly present all natural frequencies of the MIFS system. In section 3.4, the effect of the wave period on main dynamic responses of the MIFS system has been clarified, which is helpful for estimating main natural frequencies of the MIFS system (shown in Fig.7). Thanks for your kind consideration.
- L40: Delete phrase ‘and so on’.
Response: Thank you for this comment!
The corresponding description in this paper has been revised according to the comment.
- L500: Replace “locals” with “can be located”
Response: Thank you for this comment!
The corresponding description in this paper has been revised according to the comment.
References:
- Bea, R. G., R. Iversen, and T. Xu. "Wave-in-deck forces on offshore platforms." J. Offshore Mech. Arct. Eng. 123.1 (2001): 10-21.
- Douglass et al (2006). Wave forces on bridge decks. Washington, DC: U.S. Dept. of Transportation, Federal Highway Administration, Office of Bridge Technology https://www.yumpu.com/en/document/read/7198095/wave-forces-on-bridge-decks
- Allsop et al (2007). New prediction method for wave-in-deck loads on exposed piers/jetties/bridges. In Coastal Engineering 2006: (In 5 Volumes) (pp. 4482-4493).
- Xiang et al. (2020). Tsunami Loads on a Representative Coastal Bridge Deck: Experimental Study and Validation of Design Equations. Journal of Waterway, Port, Coastal, and Ocean Engineering, 146(5), 04020022. https://doi.org/10.1061/(ASCE)WW.1943-5460.0000560
- Istrati et al (2018): “Deciphering the tsunami wave impact and associated connection forces in open-girder coastal bridges”, Journal of Marine Science and Engineering, 2018, MDPI, 6 (148).
- Anagnostopoulos, S. A. (1982). Dynamic response of offshore platforms to extreme waves including fluid-structure interaction. Engineering Structures, 4(3), 179-185. https://doi.org/10.1016/0141-0296(82)90007-4
- Istrati et al (2016). Large-scale experiments of tsunami impact forces on bridges: The role of fluid-structure interaction and air-venting. In The 26th International Ocean and Polar Engineering Conference. OnePetro.https://onepetro.org/ISOPEIOPEC/proceedings-abstract/ISOPE16/All-ISOPE16/ISOPE-I-16-456/16865
- Choi, S. J., Lee, K. H., & Gudmestad, O. T. (2015). The effect of dynamic amplification due to a structure׳ s vibration on breaking wave impact. Ocean Engineering, 96, 8-20. https://doi.org/10.1016/j.oceaneng.2014.11.012
- Istrati D, Buckle IG (2014): Effect of fluid-structure interaction on connection forces in bridges due to tsunami loads. Proceedings of the 30th US-Japan Bridge Engineering Workshop, Washington DC, United States. https://www.pwri.go.jp/eng/ujnr/tc/g/pdf/30/30-10-2_Buckle.pdf
- Xiang et al (2021): Assessment of Extreme Wave Impact on Coastal Decks with Different Geometries via the Arbitrary Lagrangian-Eulerian Method. Journal of Marine Science and Engineering, 2021, MDPI, 9(12), 1342; https://doi.org/10.3390/jmse9121342
- Peregrine et al (2004). “Violent water wave impact on walls and the role of air,” Proceedings of the 29th International Conference on Coastal Engineering, 1(4), 4005–4017
- Istrati, D., & Buckle, I. (2019). Role of trapped air on the tsunami-induced transient loads and response of coastal bridges. Geosciences, 9(4), 191, https://doi.org/10.3390/geosciences9040191
- Van der Meer et al (2005). Wave transmission and reflection at low-crested structures: Design formulae, oblique wave attack and spectral change. Coastal Engineering, 52(10-11), 915-929.
- Rudmana, M., & Cleary, P. W. (2009). Oblique impact of rogue waves on a floating platform. In The Nineteenth International Offshore and Polar Engineering Conference. OnePetro.
- Istrati, D., Buckle, I.G. (2021): Tsunami Loads on Straight and Skewed Bridges–Part 2: Numerical Investigation and Design Recommendations (No. FHWA-OR-RD-21-13). Oregon. Dept. of Transportation. Research Section, https://rosap.ntl.bts.gov/view/dot/55947
- Collins et al (2018). Directional wave spectra observed during intense tropical cyclones. Journal of Geophysical Research: Oceans, 123(2), 773-793. https://doi.org/10.1002/2017JC012943
Special thanks for your valuable time and wonderful comments.
Sincerely yours,
Best regards,
Nianixn Ren

Reviewer 2 Report
See attached PDF.

Author Response
Dear editors and reviewers:
We sincerely thank the editor and all reviewers for their valuable feedback, which are very helpful to improve the quality of our manuscript. The reviewers’ comments are laid out below in blue, and our responses are given in black. Revised portions are marked in red in the revised manuscript.
Detailed responses to the comments of the reviewer:
Major comments: The manuscript presents the numerical modelling of a modular integrated floating structure system constrained by a dolphin-fender mooring. The objectives of the work are clear, and the methodology is correct. The text does not present major flaws, but I have several questions and suggestions that will improve the paper. My suggestion to the Editor is "major review" due to the large number of comments.
Response: Thank you for your positive comment and valuable time!
Minor comments:
- The proposed moorings suffer from bending moments in the connection between the monopile and the sea bottom. These stresses will be a major problem, especially in VLFS. Is this a feasible approach? The authors should comment on this issue.
Response: Thank you for this comment!
Just as the reviewer said, the horizontal force acting on the monopole will result in challenging bending moments. In order to mitigate the modules’ horizontal force acting on the monopole, the horizontal WEC device has been added between the monopile and it adjacent module, which can using their relative horizontal motion to produce energy (in updated Figure 1a). Therefore, in the view of energy conservation, the horizontal force (as well as bending moments) acting on the monopile of the MFS system may be effectively reduced to some degree.
The corresponding description has been revised according to this comment.
Figure 1a
- Since the monopiles are assumed to be rigid, the authors are overestimating the forces. Comment that on the text.
Response: Thank you for this comment!
Just as the reviewer said, the loads acting on the monopole tend to be overestimated to some degree, due to not consider both the structure deformation of the monopole and its soil-structure interaction effect. Considering this work is mainly for the conceptual design of the proposed MIFS system, the monopile in this work is preliminarily simplified as a rigid body (similar with other references), and it may be convenient for further comparing the numerical results with corresponding model test data.
The corresponding description has been revised according to this comment.
- Kim, J.-H.; Hong, S.-Y.; Cho, S.-k.; Park, S.-H. In Experimental Study On a Dolphin-Fender Moored Pontoon-Type Structure, The Fourteenth International Offshore and Polar Engineering Conference, OnePetro: 2004.
- Cho, K.-N., A Study on the Design of Dolphin System for VLFS. Journal of the Computational Structural Engineering Institute of Korea 2006, 19 (1), 105-111.
- Section 2.1 - The components of Fig. 1 should be numbered and referenced in the text by their number. Figure 1 a) and b) must be placed on the same page.
Response: Thank you for this comment!
The Figure 1 has been revised according to the comment.
(a)
(b)
Figure 1. Conceptual sketch of the MIFS with DFM: (a) Side view; (b) Top view.
- The (capital) "K" denotes degrees Kelvin. It must be replaced by "k", the prefix of kilo.
Response: Thank you for this comment!
The corresponding description has been corrected according to this comment.
- Subscripts that are not variables must be written in upright characters and not in italics. Italics are reserved for variables like "i" and "j".
Response: Thank you for this comment!
The corresponding description has been corrected according to this comment.
- The value of the viscous damping Ci is not defined. Please present and justify the value used in the computations.
Response: Thank you for this comment!
Considering the accurate viscous damping of the MIFS system is really challenging to be well estimated, it usually needs further model test data. Therefore, we haven’t considered vicious damping in this work. The Ci in Eq. 1 can be an artificial damping used to compensate for viscous flow effects obtained in model tests, but it has been simplified to be zero in this work.
- Eq. (3) is unreadable.
Response: Thank you for this comment!
The corresponding description of Eq. (3) has been revised according to this comment.
- Present details about the time integration and how the discontinuity of the bottom fender impact force was dealt with in the formulation.
Response: Thank you for this comment!
In Eq. (4) and Figure 2, it can be seen that if the relative bottom “negative” horizontal motion between two adjacent modules is less than their gap (2m), the bottoms of the two adjacent modules will impact. Then the impact force will be observed by their bottom fender elements, and the value of the impact force can be simplified as the description in Eq. (4). In fact, for the safety of the MIFS system, the bottom-impact accident among modules may be not allowed in practice. Therefore, we preliminarily optimized the parameters of the MIFS system to successfully avoid the bottom impact accident. In other words, there is no observed bottom impact force from bottom fender elements in all test cases. Thanks for your kind consideration.
- What are the red half-circles present in Fig. 2?
Response: Thank you for this comment!
The red half-circles present in Fig. 2 are the bottom fender elements to monitor potential bottom impact force between adjacent modules.
- Eq. (7) - Are the velocity potentials a function of time and position or just time?
Response: Thank you for this comment!
The velocity potentials are functions of time, as well as with the information of position. More information can refer to the following references.
- Ren, N. X.; Wu, H. B.; Ma, Z.; Ou, J. P. Hydrodynamic analysis of a novel modular floating structure system with central tension-leg platforms. Ships Offshore Struc. 2020, 15 (9), 1011-1022.
- Ren, N. X.; Wu, H. B.; Liu, K.; Zhou, D. C.; Ou, J. P. Hydrodynamic Analysis of a Modular Floating Structure with Tension-Leg Platforms and Wave Energy Converters. Mar. Sci. Eng. 2021, 9 (4), 424.
- Ren, N. X.; Zhang, C.; Magee, A. R.; Hellan, Ø.; Dai, J.; Ang, K. K., Hydrodynamic analysis of a modular multi-purpose floating structure system with different outermost connector types. Ocean Eng. 2019, 176, 158-168.
- The authors must plot the hydrodynamic coefficients of all the bodies as a function of the frequency and for each degree of freedom. The results must be reproducible by other authors. I will suggest the rejection of the paper if this information is omitted.
Response: Thank you for this comment!
Considering the MIFS system involving many modules with complex hydrodynamic and mechanical interaction effects (refer to Eq. (1) ~ (7)), it may be challenging to perfectly present all natural frequencies of the MIFS system. In section 3.4, the effect of the wave period on main dynamic responses of the MIFS system has been clarified, which may be helpful for estimating main natural frequencies of the MIFS system (shown in Fig.7). Thanks for your kind consideration.
- How was the radiation damping computed in the time domain?
Response: Thank you for this comment!
The AQWA code is widely-used for the hydrodynamic interaction analysis of modular floating systems. Some related references are listed as follows (our previous works), indicating that it is feasible and reliable for the hydrodynamic analysis of modular floating systems (including the radiation damping calculation). Most of the numerical results based on AQWA are in good agreement with corresponding model test data. However, considering physical model tests are essential for validating the numerical results and improving numerical method, so we have been planning a model test at the end of this year. Then, we would like to compare the obtained test data with the corresponding numerical results, to prepare a new paper for submitting to the journal. Thanks for your kind consideration.
- Ren, N.X., Zhang, C., Magee, A. R., et al., 2019. Hydrodynamic analysis of a modular multi-purpose floating structure system with different outermost connector types. Ocean Engineering.176, 158-168.
- Ren, N.X., Wu, H., Ma, Z., et al., 2019. Hydrodynamic analysis of a novel modular floating structure system with central tension-leg platforms. Ships and Offshore Structures. Ships and Offshore Structures.15 (9), 1011-1022.
- Ren, N.X., Ma, Z., Shan, B.H., et al., 2020. Experimental and numerical study of dynamic responses of a new combined TLP type floating wind turbine and a wave energy converter under operational conditions. Renewable Energy. Ships and Offshore Structures. 151,966-974.
- Ren, N.X., et al., 2018. Experimental and numerical study of hydrodynamic responses of a new combined monopile wind turbine and a heave-type wave energy converter under typical operational conditions. Ocean Engineering. 159,1-8.
- Liu, Y.Q., Ren, N.X., Ou, J.P., 2021. Hydrodynamic analysis of a hybrid modular floating structure system and its expansibility. Ships and Offshore Structures. DOI:10.1080/17445302.2021.1996110.
- Yanwei Li, Nianxin Ren, Xiang Li*, Jinping Ou. Hydrodynamic Analysis of a novel modular floating structure system integrated with floating artificial reefs and wave energy converters. Journal of Marine Science and Engineering, 2022, 10, 1091. https://doi.org/10.3390/jmse10081091
- Legend of Figure 3 a) - replace "Dployment" with "Deployment". Caption Figures must be placed below the Figures and not on the next page.
Response: Thank you for this comment!
The corresponding description has been corrected according to this comment.
- In Figure 4 and others, the unit of the "y" axis should be "(º)" and not "(º )" (an extra space).
Response: Thank you for this comment!
The corresponding description has been corrected according to this comment.
- Table 2 and elsewhere, remove the dot between units, ex N·m.
Response: Thank you for this comment!
The corresponding description has been corrected according to this comment.
- Figure 5 and elsewhere, remove the extra space in the numbers appearing on the "x" axis.
Response: Thank you for this comment!
The corresponding Figures have been corrected according to this comment.
- Across the text, use a small space between the number and the units.
Response: Thank you for this comment!
The corresponding description has been corrected according to this comment.
- It is unclear whether the results presented are for regular waves or spectra. Please clarify.
Response: Thank you for this comment!
The numerical results in Section 3.1 ~3.4 and Section 4.1~4.2 are for typical regular wave conditions to investigate key parameters’ effects on main dynamic responses of the MIFS system, while the numerical results in Section 3.5 and Section 4.3 are for typical extreme irregular wave conditions to check the safety (extreme responses) of the MIFS system.
The corresponding description has been corrected according to this comment.
- Describe the expected wave climate of the deployment site. Why the authors selected periods between 4 and 14 seconds to report the results?
Response: Thank you for this comment!
Considering the MIFS system usually designs to work near the artificial or natural shelters (islands or reefs) in a mild sea zone, the wave conditions for MIFS systems are not very serious (usually Hs< 4m, with the wave period less than 14s). The wave conditions (for one certain zone in South China Sea) selected in this paper are mainly based on our previous work, and more information can refer to the following references.
- Ren, N.X., Wu, H., Ma, Z., et al., 2019. Hydrodynamic analysis of a novel modular floating structure system with central tension-leg platforms. Ships and Offshore Structures. Ships and Offshore Structures.15 (9), 1011-1022.
- Ren, N.X., Ma, Z., Shan, B.H., et al., 2020. Experimental and numerical study of dynamic responses of a new combined TLP type floating wind turbine and a wave energy converter under operational conditions. Renewable Energy. Ships and Offshore Structures. 151,966-974.
- Ren, N.X., et al., 2018. Experimental and numerical study of hydrodynamic responses of a new combined monopile wind turbine and a heave-type wave energy converter under typical operational conditions. Ocean Engineering. 159,1-8.
- Liu, Y.Q., Ren, N.X., Ou, J.P., 2021. Hydrodynamic analysis of a hybrid modular floating structure system and its expansibility. Ships and Offshore Structures. DOI:10.1080/17445302.2021.1996110.
- Yanwei Li, Nianxin Ren, Xiang Li*, Jinping Ou. Hydrodynamic Analysis of a novel modular floating structure system integrated with floating artificial reefs and wave energy converters. Journal of Marine Science and Engineering, 2022, 10, 1091. https://doi.org/10.3390/jmse10081091
- Figure 12 a) and elsewhere, why not compute the results for larger periods?
Response: Thank you for this comment!
Considering the MIFS system usually designs to work near the artificial or natural shelters (islands or reefs) in a mild sea zone, the wave conditions for MIFS systems are not very serious (usually Hs< 4m, with the wave period less than 14s). The wave conditions (for one certain zone in South China Sea) selected in this paper are mainly based on our previous work, and more information can refer to related references.
- In the plots Use MW instead of kW.
Response: Thank you for this comment!
The corresponding description has been revised according to this comment.
- The abbreviations should be sorted by the abbreviation and not by the description.
Response: Thank you for this comment!
The corresponding description has been reordered according to this comment.
- Further comments can be found in the annotated PDF.
Response: Thank you for this comment!
The corresponding description has been revised according to all comments in the annotated PDF.
Thanks for your valuable time and kind consideration!
Sincerely yours,
Best regards,
Nianixn Ren

Reviewer 3 Report
The paper is easy to read and results discussion is very good. It is only necessary to review some sentences and perhaps improve some figures.

Author Response
Dear reviewer,
We sincerely thank your positive comments and valuable feedbacks, which we have used to improve the quality of our manuscript. The reviewer's comments list in the attached PDF have been directly answered in corresponding PDF file. Revised portions are marked in red in the revised manuscript.
Thanks for your valuable time and kind consideration!
Sincerely yours,
Best regards,
Nianixn Ren

Reviewer 4 Report
Please see the attachment.

Author Response
Dear editors and reviewers:
We sincerely thank the editor and all reviewers for their valuable feedback, which are very helpful to improve the quality of our manuscript. The reviewers’ comments are laid out below in blue, and our responses are given in black. Revised portions are marked in red in the revised manuscript.
Response to the comments of reviewer #3:
The present paper proposed a modular integrated floating structure (MIFS) system with tidal self-adaptation dolphin-fender mooring (DFM) which could be used as an anti-motion system. Further, the hydrodynamic interaction effect and the connectors as well as the dynamic responses of the modular integrated floating structure (MIFS) system and the WEC’s output power characteristics are also investigated. The preset work is of recent interest in the field of wave structure interaction problems that cover a broad area of application. However, this reviewer writes some comments and suggestions that need to be addressed in the paper to improve the quality and access to wide readability.
Response: Thank you for your positive comment!
- Introduction: The present manuscript needs to include some recent works carried out by other researchers.
Response: Thank you for this comment!
More recent references have been added in “Introduction”.
- Mohapatra S. C. and Soares C. G. Hydroelastic Response to Oblique Wave Incidence on a Floating Plate with a Submerged Perforated Base. Journal of Marine Science and Engineering, 2022, 10, 1205. https://doi.org/10.3390/jmse10091205
- Cheng Y., Xi C., DaiS. S. , Ji C.Y., Collu M., Li M. X., Yuan Z. M., Incecik A. Wave Energy extraction and hydroelastic response reduction of modular floating breakwaters as array wave energy converters integrated into a very large floating structure. Applied Energy 2022; 306: 117953.
- Li Y. W., Ren N. X., Li X., Ou J. P. Hydrodynamic Analysis of a novel modular floating structure system integrated with floating artificial reefs and wave energy converters. Journal of Marine Science and Engineering, 2022, 10, 1091. https://doi.org/10.3390/jmse10081091
- The symbols used in Equations (3-4) are not consistent.
Response: Thank you for this comment! The terms in Equation (3-4) are of a bit difference, so the symbols are a bit different. Thanks for your kind consideration.
- Throughout the paper there are missing “periods” and “commas” at the end of the Equations.
Response: Thank you for this comment!
Considering the equations in this paper are all in independent rows, so “periods” and “commas” are omitted at the end of the equations.
- Section 3: To make the present results interesting and check the level of accuracy of the present model, it is recommended that the authors could add some comparison results between the present and the existing published results (in a specific case) would be nice (as there is no independent Exp. tests results and other methodology are documented, however, the authors have mentioned it is one of the future tasks in the conclusion).
Response: Thank you for this comment!
Just as the reviewer said, numerical results of this work need further validation for a better understanding. In fact, the AQWA code is widely-used for the hydrodynamic analysis of multi-body floating systems. There are some existing references listed as follows (our previous works), indicating that it is feasible and reliable for the hydrodynamic analysis of modular floating systems. Most of the numerical results based on AQWA are in good agreement with corresponding model test data. However, considering physical model tests are essential for validating the numerical results and promoting numerical method, so we have been planning a model test at the end of this year. Then, we would like to compare the obtained test data with the numerical results in this paper, to prepare a new paper for submitting to JMSE.
Thanks for your kind consideration.
- Ren, N.X., Zhang, C., Magee, A. R., et al., 2019. Hydrodynamic analysis of a modular multi-purpose floating structure system with different outermost connector types. Ocean Engineering.176, 158-168.
- Ren, N.X., Wu, H., Ma, Z., et al., 2019. Hydrodynamic analysis of a novel modular floating structure system with central tension-leg platforms. Ships and Offshore Structures. Ships and Offshore Structures.15 (9),1011-1022.
- Ren, N.X., Ma, Z., Shan, B.H., et al., 2020. Experimental and numerical study of dynamic responses of a new combined TLP type floating wind turbine and a wave energy converter under operational conditions. Renewable Energy. Ships and Offshore Structures. 151,966-974.
- Ren, N.X., et al., 2018. Experimental and numerical study of hydrodynamic responses of a new combined monopile wind turbine and a heave-type wave energy converter under typical operational conditions. Ocean Engineering. 159,1-8.
- Liu, Y.Q., Ren, N.X., Ou, J.P., 2021. Hydrodynamic analysis of a hybrid modular floating structure system and its expansibility. Ships and Offshore Structures. DOI:10.1080/17445302.2021.1996110.
- Yanwei Li, Nianxin Ren, Xiang Li*, Jinping Ou. Hydrodynamic Analysis of a novel modular floating structure system integrated with floating artificial reefs and wave energy converters. Journal of Marine Science and Engineering, 2022, 10, 1091. https://doi.org/10.3390/jmse10081091
- As the present study focuses on the expansion of the floating structure acting as an anti-motion and WEC which includes mooring lines, then this reviewer highly recommends considering the following recent development and other references from the literature as well. https://doi.org/10.3390/jmse10091205
Response: Thank you for this comment!
The corresponding reference has been added in the paper.
- Mohapatra S. C. and Soares C. G. Hydroelastic Response to Oblique Wave Incidence on aFloating Plate with a Submerged Perforated Base. Journal of Marine Science and Engineering, 2022, 10, 1205. https://doi.org/10.3390/jmse10091205
- Conclusion: Could you please elaborate a little bit more on this statement “These challenging works should be investigated in future work.”
Response: Thank you for this comment!
The corresponding description has been revised according to this comment.
Thanks for your valuable time and kind consideration!
Sincerely yours,
Best regards,
Nianixn Ren
Round 2
Reviewer 1 Report
My comments have been properly addressed in the revised version.
Thank you.
Author Response
Dear the reviewer:
Thanks for your valuable time and kind consideration!
Sincerely yours,
Best regards,
Nianixn Ren
Reviewer 4 Report
The detailed clarification provided by the authors is acceptable and this reviewer believes that the authors will follow up on the comparison results in their future works.
Author Response

(The authors gave the same response as above.)
